# Condensate-forming eIF4ET ensures adequate levels of meiotic proteins to support oocyte storage

Priyankaa Bhatia[1], Judith Tafur[1], Ruchi Amin[2], Nicole E Familiari[1], Kan Yaguchi[1], Vanna M Tran[1], Alec Bond[1], Orhan Bukulmez[2], Jeffrey B Woodruff[1]

**Animals store oocytes in a dormant state for weeks to decades before ovulation. The homeostatic programs that oocytes use to endure long-term storage are poorly understood. Using female nematodes as a short-lived model, we found that oocyte formation and storage required IFET-1, the conserved eIF4E-transporter protein (eIF4ET). IFET-1 co-assembled with CAR-1 (Lsm14) to form micron-scale condensates in stored oocytes, which dissipated after oocyte activation. Depletion of IFET-1 destabilized the stored oocyte proteome, leading to lower translation, a decline in microtubule maintenance proteins, and errors in microtubule organization and meiotic spindle assembly. Deleting domains within IFET-1 impaired oocyte storage without affecting oocyte formation. Thus, in addition to establishing a healthy oocyte reserve in young mothers, IFET-1 ensures that correct levels of cytoskeletal proteins are maintained as oocytes age. eIF4ET also localized to micron-scale puncta in dormant human oocytes. Our results clarify how eIF4ET maintains the oocyte reserve and further support eIF4ET dysfunction as an upstream cause of embryonic aneuploidy and age-related infertility.**

## Introduction

The survival of many animal species depends on their ability to store high-quality oocytes for prolonged periods of time. Such storage typically involves cell cycle arrest of the oocyte, typically in meiotic prophase I. In mammals, stored oocytes are small (~25–35 $\mu m$ in diameter) and exist in a primordial follicle, where they are supported by squamous pre-granulosa cells (1, 2). These oocytes can be reactivated to grow into immature GV oocytes (~100 $\mu m$ in diameter), which then are either destroyed (atresia) or mature further into a metaphase-arrested form that is released for fertilization (ovulation) (1, 2). A woman's pool of stored oocytes, called the ovarian reserve, declines immediately after birth, even before ovulation can occur. This decline continues as the woman

ages, ultimately leading to a depleted ovarian reserve and menopause when she reaches 40–50 yr of age (3, 4). Oocyte quality also decreases during long-term storage, as defined by an increase in mitochondrial dysfunction, DNA damage, and meiotic spindle errors (5, 6, 7). These age-related errors are correlated with an increased risk of chromosomal abnormalities in the developing fetus and miscarriage (8). Similar phenomena have been described in a wide variety of animals (9). The confluence of declining oocyte numbers and quality are thus major contributors to age-related infertility in female mammals, yet the upstream molecular causes remain poorly understood.

Although much research has revealed mechanisms of oocyte production, cell cycle arrest at the GV stage, and post-GV maturation (1, 10), comparatively little is known about long-term oocyte maintenance. In preparation for storage, oocytes enter a state of cellular hibernation in part by reducing transcription and mitochondrial production of ATP (11, 12). Understanding storage requires identifying the molecules and mechanisms that continuously provide "life support" for the oocyte during its storage. A potential inroad is through studying a disease that affects oocyte storage. Primary ovarian insufficiency (POI) is characterized by an accelerated decline in the ovarian reserve or general dysfunction of ovarian follicles. Patients with POI typically experience irregular menstrual cycles, low anti-Müllerian hormone, and premature menopause (26–32 yr of age) (13). As these patients start out with a large ovarian reserve and can bear children early (before menopause), this disease represents a case where the body fails to preserve oocytes over time. Mutations in Eukaryotic Translation Initiation Factor 4E Nuclear Import Factor 1 (*EIF4ENIF1*), which encodes a protein from the conserved eIF4ET family, are associated with POI, suggesting a link to oocyte quality control (14, 15, 16). Consistent with this theory, EIF4ENIF1 heterozygous knockout mice display reduced follicle pools and subfertility, recapitulating key features of POI (17). In somatic tissue culture cells, EIF4ENIF1 binds to and mediates nuclear import of the mRNA cap-binding protein eIF4E (18, 19, 20). Through the use of nonnatural reporters, these studies concluded that EIF4ENIF1 associates with P-bodies to inhibit protein translation and stabilize mRNAs. Injection of EIF4ENIF1

[1]Department of Cell Biology, Department of Biophysics, UT Southwestern Medical Center, Dallas, TX, USA   [2]Division of Reproductive Endocrinology and Infertility, Department of Obstetrics and Gynecology, UT Southwestern Medical Center, Dallas, TX, USA

Correspondence: Jeffrey.woodruff@utsouthwestern.edu

into frog GV oocytes also repressed translation of a luciferase reporter [21]. EIF4ENIF1 is found in P-body–like granules in dormant mouse oocytes [17] and in mitochondria-associated granules (MARDO) in mammalian GV oocytes [22]. However, it is not known whether EIF4ENIF1 is required specifically during oocyte storage and whether it regulates the global pool of natural mRNAs and proteins in that context.

The mechanism of oocyte storage remains poorly understood because of the difficulty in accessing dormant oocytes. In mammals, dormant oocytes are obtained only through invasive surgical resection of the ovarian cortex. In vitro gametogenesis also cannot be used because it does not yet recapitulate formation and long-term maintenance of the primordial follicle [23], the fundamental storage unit that comprises the dormant oocyte surrounded by pre-granulosa cells [1]. Thus, it is advantageous to use a genetic model organism with a short lifespan and oocytes that are easily visualized in situ.

Here, we use *C. elegans* females to study oocyte maintenance mechanisms. Most nematodes are gonochoric and store immature oocytes arrested in meiotic prophase I. *C. elegans* is an outlier in that it recently evolved hermaphroditism, yet *C. elegans* females retain the ability to store arrested oocytes in cases where sperm are depleted naturally or genetically (*fog-2* or *fem-1* mutations) [24, 25]. Thus, oocyte storage is an ancestral feature of nematodes. As *Caenorhabditis* species live for several weeks and store their oocytes for less time, they represent an attractive short-lived model for oocyte storage.

## Results

### Dysfunction of the eIF4ET homolog (IFET-1) impairs oocyte formation and storage in female nematodes

We developed an assay to test the effects of storage time on oocyte quality. We generated a population of genetically identical *C. elegans* females through the use of a temperature-sensitive *fem-1(hc17)* loss-of-function allele. After synchronizing their development, all worms reached adulthood and formed arrested oocytes at the same time (first day of adulthood = day 1). We then split the females into cohorts, letting them wait another 1–5 d before introducing young males for mating. At the first day of mating (day 2), the female worms contained, on average, 34 ± 7 oocytes (n= 20). Worms were allowed to mate and lay fertilized eggs over 12–16 h before removing all adults from the plates. Because females can produce new oocytes after ovulating their stored oocytes, we reported on worms that laid 34 or fewer eggs. After 2 d, we measured the number of hatchlings divided by the number of eggs laid (oocyte viability) (Fig 1A). Under these conditions, average oocyte viability declined with storage time, reaching 63.4% by day 6 (Fig 1B). From these data, we infer that *C. elegans* oocyte quality declines in proportion to their storage time, like what is seen in mammals.

IFET-1 is the *C. elegans* homolog of EIF4ENIF1, but its role during oocyte storage has not been tested. Full depletion of IFET-1 using RNAi rendered almost all stored oocytes inviable (Figs 1B and S1A); IFET-1 levels remained low in stored oocytes, even after worms were

removed from the RNAi feeding plates (Fig S1B). IFET-1 depletion also impaired ovulation in females and oocyte viability in hermaphrodites, which do not arrest their oocytes, consistent with a previous study [26]. Thus, full knockdown of IFET-1 has pleiotropic effects making it difficult to assess its role specifically during oocyte storage. Patients with EIF4ENIF1-linked POI all have heterozygous mutations, implying that failed oocyte storage could be caused by partial reduction of EIF4ENIF1 activity. In our system, partial knockdown of IFET-1 allowed young mothers to bear healthy progeny, although initial viability was reduced to 31%. Oocyte viability declined faster than in controls, reaching zero after 4 d of storage (Fig 1B). Oocytes in *ifet-1(RNAi)* females were irregular in shape, as expected; however, all but the most proximal oocyte maintained arrest in diakinesis, as indicated by the presence of intact germinal vesicles and condensed chromosomes (Fig S1C). Thus, oocytes are still arrested in the absence of IFET-1, but they are largely inviable. These results support previous findings that IFET-1 is required for oocyte formation [26], and they further suggest that IFET-1 could be required for maintenance of stored oocytes.

To definitively test the necessity of IFET-1 for the storage of healthy oocytes, we sought a way to deplete IFET-1 only after oocyte formation is complete. Thus, we generated a *fem-1* strain expressing a version of IFET-1 that is degraded by the addition of auxin (IFET-1::mScarlet::AID*). As outlined in Fig 1C, we placed synchronized adult females, which already have a row of arrested oocytes (day 1) on plates with or without 4 mM auxin for 24 h at 20°C. Then, we allowed worms to recover on normal Nematode Growth Media (NGM) plates for 8 h at 20°C and allowed mating with males and egg laying for 12–16 h at 16°C (day 3). On average, worms laid 26 (control) versus 13 (auxin) eggs during the laying period (n = 34 versus 28 mothers, respectively); thus, any eggs or larvae on the plate derived from preformed, healthy oocytes. Finally, we measured oocyte viability using the same method described above.

Auxin treatment depleted the IFET-1::mScarlet::AID* signal relative to controls (Fig 1D and E). In both cases, oocyte morphology appeared normal, suggesting that oocytes were formed properly (Fig S1D). After 8 h of post-auxin removal, just before mating, the IFET-1 signal did not recover within oocytes but did increase in the gonads (Fig S1E). Thus, auxin-induced depletion of IFET-1::mScarlet::AID* in oocytes is robust and IFET-1 does not re-accumulate in stored oocytes after auxin removal. After mating, we observed that auxin treatment significantly reduced oocyte viability (Fig 1F).

We considered the possibility that this phenotype could be due to IFET-1 being essential during embryogenesis. However, treatment of excised embryos with 1 mM auxin rapidly depleted IFET-1 but had no detectable effect on embryo development and viability (Fig S1F and G). Auxin treatment did not significantly affect the fertility of control females not expressing the degron, indicating that auxin is not generally toxic (Fig S1H). Thus, the oocyte viability phenotype is due to lack of IFET-1 specifically during oocyte storage. We conclude that IFET-1 is required to maintain oocyte quality during days-long storage.

### IFET-1 localizes to condensates that do not contain mitochondria

To understand the function of IFET-1 in stored oocytes, we used confocal microscopy to visualize IFET-1 tagged with mMaple (IFET-1::

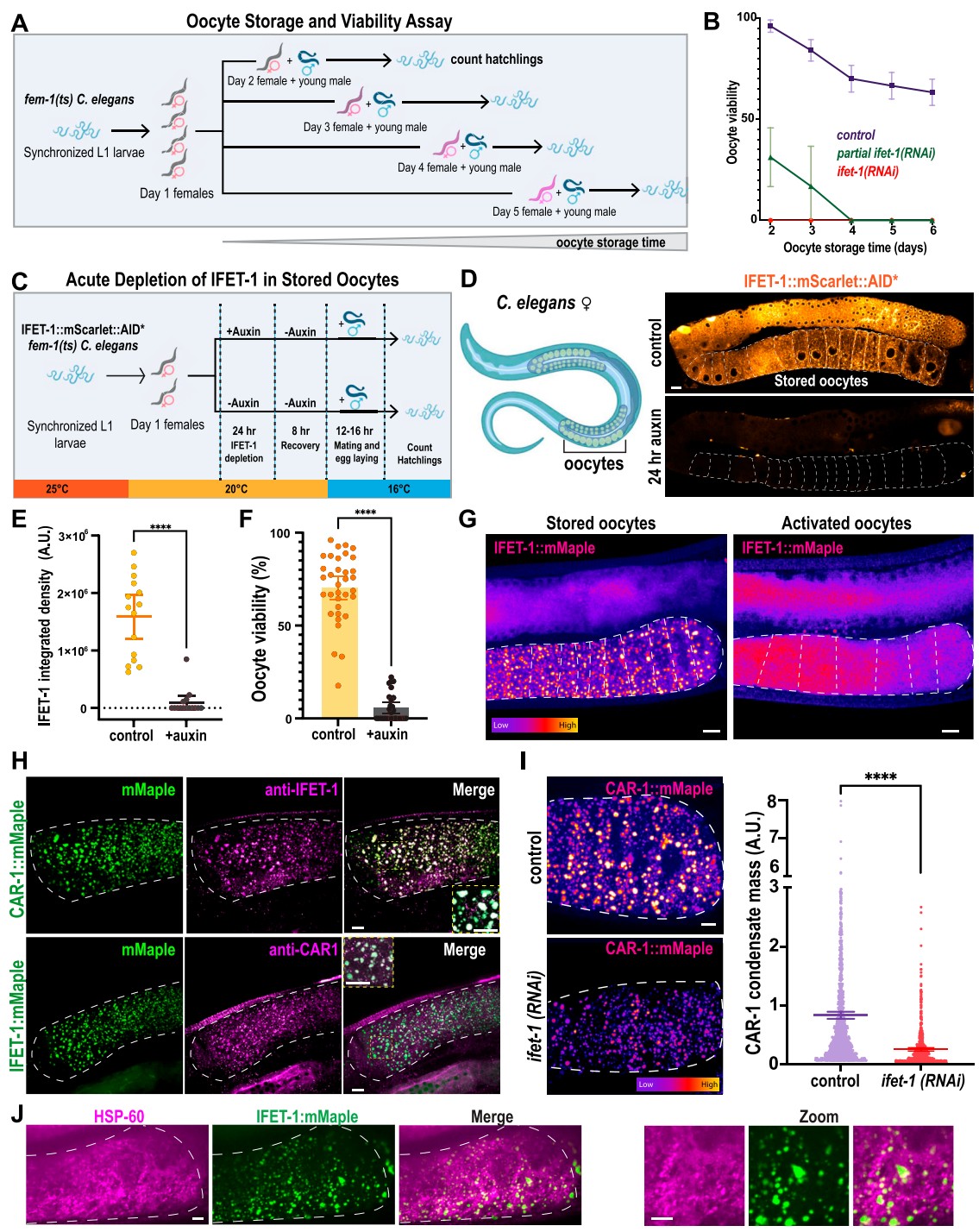

**Figure 1. IFET-1 is required for long-term storage of quiescent oocytes in *C. elegans* females.**
**(A)** Strategy to study oocyte storage and viability in *C. elegans* females. *fem-1(hc17)* worms were synchronized at L1, grown to adulthood at 25°C to induce feminization and oocyte storage, and then separated into five different cohorts. Female cohorts were mated with young males on different days, and then, offspring were analyzed. **(A, B)** Percentage of oocytes that hatched, as described in (A) (mean ± 95% C.I.; n = 12–30 worms, brood sizes ≤34 eggs laid). Partial knockdown of *ifet-1* was achieved by feeding on RNAi plates for 48 h; full knockdown was achieved by feeding for 72 h. See the Supplemental Data 3 for all viability data. **(C)** Strategy to deplete IFET-1 in stored oocytes after their formation. Synchronized *fem-1(hc17) ifet-1::mScarlet::AID\* TIR-1::T2A::BFP* adult female worms were placed on plates with 0 or 4 mM auxin for 24 h. Females were removed from auxin for 8 h before mating with males. **(D)** Left, diagram of a *C. elegans* female. Right, confocal image of day 3 adult females expressing IFET-1::mScarlet::AID\* grown on plates containing 0 mM (control) or 4 mM auxin. Scale bar, 10 μm. **(D, E)** Integrated fluorescence intensity of IFET-1::mScarlet::AID\* signal in stored oocytes from panel (D) (mean ± 95% C.I.; n = 15 worms). **(F)** Viability of stored oocytes after mating in IFET-1::mScarlet worms with or without auxin treatment (mean ± 95% C.I.; n = 34 [control] versus 28 [auxin] worms, brood sizes =26 [control] versus 13 [auxin] eggs per mother). **(G)** Live-cell confocal images of IFET-1::mMaple in stored (female, not mated) and activated (hermaphrodite) oocytes. Scale bars, 10 μm. **(H)** Immunofluorescence images of stored oocytes. Scale bars, 10 μm. **(I)** Left, images of

mMaple). Worms expressing IFET-1::mMaple were viable and fertile (Fig S1I). In stored oocytes, IFET-1::Maple localized to numerous micron-scale droplets, which we term condensates. After activation of the oocytes through fertilization, these condensates were dissolved and the IFET-1::Maple signal was redistributed evenly throughout the oocyte cytoplasm (Fig 1G). A similar behavior was reported for CAR-1, which localizes to droplets in stored oocytes (27, 28) and interacts with IFET-1 in embryo extracts (29). To co-visualize CAR-1 and IFET-1, we tagged endogenous CAR-1 at its C terminus with mMaple and generated antibodies against both proteins for immunofluorescence. CAR-1::mMaple localized to condensates in stored oocytes, as expected, and co-localized with untagged IFET-1 (Fig 1H). IFET-1::mMaple also co-localized with untagged CAR-1. RNAi depletion of IFET-1 reduced CAR-1 localization to condensates (Fig 1I). These results indicate that IFET-1 and CAR-1 co-assemble into supramolecular condensates in stored oocytes.

We wondered whether the IFET-1/CAR-1 condensates function as Balbiani bodies, which are reported to encapsulate and confine mitochondria in the dormant oocytes of zebrafish, frogs, and humans (30, 31, 32). However, mitochondria are not spatially confined in stored *C. elegans* oocytes (33), and we saw no significant co-localization between HSP60 (mitochondrial marker) and IFET-1 condensates (Fig 1J). We conclude that IFET-1 condensates are not primitive Balbiani bodies but rather a distinct class of protein-based supramolecular assembly that exists in stored oocytes.

### Disrupting individual domains in IFET-1 perturbs oocyte storage but not formation

We next tested which domains within IFET-1 are critical for its function in oocytes. IFET-1 contains several conserved features: an eIF4E homology domain (Eif4EHD), Cup homology domain (CHD), NLS, a region rich in glutamines (PolyQ), and a C-terminal coiled-coil (Fig 2A). We used CRISPR to delete each of these regions in worms expressing endogenously tagged IFET-1::mMaple (Fig 2A). We first analyzed homozygous mutant hermaphrodite worms, which make and activate oocytes continuously with no storage phase (34). In this condition, all mutant worms produced viable oocytes; ΔCHD, ΔNLS, and ΔPolyQ mutants were nearly identical to controls, whereas the ΔCoiledcoil mutant showed slightly reduced viability (Fig 2B).

Mutant female worms (ΔCHD, ΔNLS, ΔPolyQ, ΔCoiledcoil) formed oocytes with proper cuboidal morphology but could not store them effectively: after 2 d of oocyte storage followed by mating, embryo viability was reduced compared with eggs laid by age-matched hermaphrodites (Fig 2B). IFET-1::mMaple mutants localized to condensates in stored oocytes to differing degrees (Fig 2C and D). Mutants were expressed at 55–64% of WT levels (Fig 2E). Importantly, for each mutant, expression levels were similar between female worms with stored oocytes and hermaphrodites with non-stored oocytes (Fig S2A and B); thus, the oocyte viability phenotype is due to a distinct role of IFET-1 in storage that is not required during active oogenesis. We speculate that these mutations

comprise IFET-1 function, especially the ΔCHD mutation, which had the most severe oocyte viability phenotype (21.8% viability after 2 d of storage) despite being expressed at comparable levels to other mutants. The ΔCoiledcoil mutant had the largest effect on condensate localization, suggesting that the coiled-coil domain helps IFET-1 co-assemble with CAR-1 to form condensates (Fig 2C). These results reveal that (1) IFET-1 functions can be separated based on domains, and (2) multiple domains are required for full-scale IFET-1 expression and function in stored oocytes. These data further support our conclusion that IFET-1 is required to form oocytes and maintain their quality during the storage phase.

### IFET-1 depletion causes imbalances in meiotic spindle protein levels

To determine how IFET-1 dysfunction impacts oocyte quality, we performed proteomic and transcriptomic analyses in control versus *ifet-1(RNAi)* female worms containing oocytes stored for 2 d. Because most of IFET-1 is expressed in the germline (oocytes + syncytial gonad; Fig 1), we report data derived from whole worms and follow up on specific hits with fluorescent reporters that are expressed in stored oocytes.

Depletion of IFET-1 caused disruption of the proteome. Out of 3,561 proteins detected in both control and RNAi sets, 530 proteins (14.9%) were significantly down-regulated, whereas 100 proteins (2.8%) were significantly up-regulated in *ifet-1(RNAi)* worms (Fig 3A; Supplemental Data 1). IFET-1 levels were reduced to 15% of control levels. This result contrasts with the expectation that IFET-1 functions as a broadscale repressor, which was based on non-native reporter assays (26). Transcriptomic analyses revealed few changes in mRNA levels in control versus *ifet-1(RNAi)* worms. Only 6.8% of all transcripts were significantly altered (4.5% up, 2.3% down) after depletion of IFET-1 (Fig 3B; Supplemental Data 2). However, there was little overlap between significant hits in the proteomic and transcriptomic data sets, suggesting that proteomic changes largely did not result from changes in transcript levels (Fig S3A and B).

Using Gene Ontology (GO) enrichment analysis of down-regulated proteins in *ifet-1(RNAi)* worms, we saw that the highest-ranking category included proteins related to embryo development and egg hatching (Fig S3C; Supplemental Data 4). Within this category were proteins required for meiotic spindle or microtubule function, such as katanin (MEI-1) and oocyte-specific isoforms of tubulin (TBA-1, TBA-2). Focusing on meiotic spindle proteins, we also saw down-regulation of kinesin-like protein 15 (KLP-15), TPX2 (TPXL-1), dynein light intermediate chain (DLI-1), Aurora A kinase (AIR-1), and other tubulin isoforms (TBB-1 and TBB-2) (Fig 3C). Not all microtubule-related proteins were down-regulated; for example, dynein heavy chain (DHC-1), ch-TOG (ZYG-9), and ASPM-1 were slightly up-regulated. Levels of the transcripts encoding these proteins were all slightly down-regulated in *ifet-1(RNAi)* worms (60–95% of control; Fig 3D) and showed no correlation with proteomic changes (Fig 3E). Thus, IFET-1 regulates levels

---

stored oocytes expressing CAR-1::mMaple. Right, quantification of total mass of CAR-1 condensates per worm (mean ± 95% C.I.; n = 850–900 condensates from 5 to 8 worms). **(J)** Immunofluorescence staining of mitochondria (HSP60) in stored oocytes expressing IFET-1::mMaple. Scale bars, 10 μm.

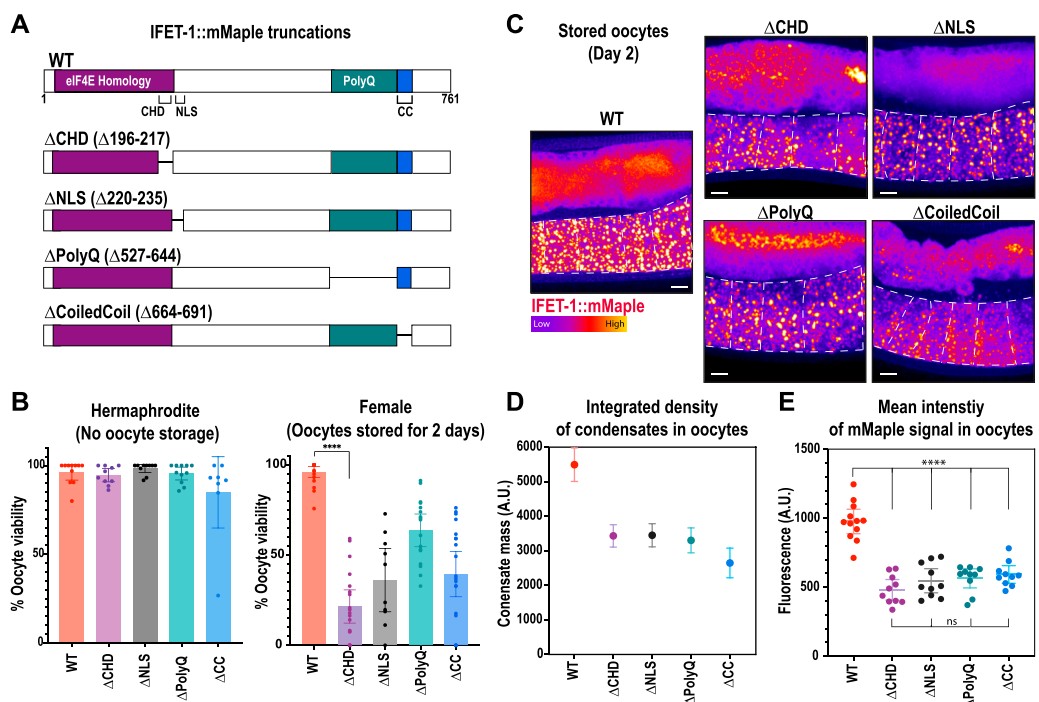

**Figure 2. Specific domains in IFET-1 contribute to oocyte storage.**
**(A)** Design of mutant constructs created with CRISPR modification of *ifet-1::mMaple* worms. Shown is the protein sequence of IFET-1 (CHD, Cup homology domain; NLS, nuclear localization signal; CC, coiled-coil domain; PolyQ, region rich in glutamines). **(B)** Quantification of oocyte viability from 2-d-old adult hermaphrodites (non-stored oocytes) and females (stored oocytes) after mating (mean ± 95% C.I.; n = embryos from 15 to 22 worms, brood size ≤34 eggs laid for female worms). See Supplemental Data 3 for raw data. **(A, C)** Confocal images of stored oocytes expressing the mutant variants of IFET-1::mMaple shown in (A). **(D)** Quantification of IFET-1::mMaple assembly into stored condensates (mean ± 95% C.I.; n = 500–600 condensates from 5 to 8 worms). **(E)** Quantification of mean IFET-1::mMaple fluorescence signal in oocytes (mean ± 95% C.I. from 10 to 15 worms).

of meiotic spindle and microtubule-related proteins in a post-transcriptional manner. It is possible that these proteomic changes are, in part, due to defects in the production of full-sized, mature oocytes, which have been reported for worms lacking functional IFET-1(26). However, proteomic analysis also revealed lower MEI-1, AIR-1, DLI-1, and TBB-1 levels in *ifet-1(ΔCHD)* mutant females, which are defective for oocyte storage and not active oogenesis (Fig S3D).

We then analyzed global levels of translation in whole worms using a SUnSET assay. Puromycin incorporation was reduced in worms containing arrested oocytes versus active oocytes (Fig S3E); thus, reduced translation is a hallmark of oocyte storage in *C. elegans*. In worms containing stored oocytes, depletion of IFET-1 further reduced puromycin incorporation compared with controls (Fig 3F). Our results suggest that lowered meiotic protein levels in *ifet-1(RNAi)* female worms are in part due to reduced translation.

## Loss of IFET-1 disrupts microtubule organization in stored and activated oocytes

Our proteomic and transcriptomic data indicated that several meiotic spindle proteins, but not their mRNAs, were down-regulated in *ifet-1* mutants. We hypothesized that such proteostatic dysfunction should affect meiotic progression, which could explain how IFET-1 loss impairs oocyte viability. Thus, we generated feminized worms expressing GFP-labeled beta tubulin (*fem-1ᵗˢ* GFP::

TBB-2), allowed them to store oocytes, and then mated them with males to induce oocyte activation and meiotic spindle assembly (Fig 4A). In control arrested oocytes (no RNAi), microtubules appeared throughout the cytoplasm, but they concentrated in bright networks near the oocyte periphery, as reported before (35). After oocytes were activated by mating, the cytoplasmic microtubules were disassembled, and the meiotic spindle was formed (Fig 4B). In *ifet-1(RNAi)* oocytes, GFP::TBB-2 did not localize to bright peripheral networks and overall expression was lower, consistent with our proteomic data (Fig 4C and D). After mating, bipolar formation of a single meiotic spindle per oocyte failed; instead, we observed diminutive spindle-like structures, microtubule clusters, or aggregates of tubulin signal (Fig 4C). Thus, IFET-1 is needed for organization of the microtubule cytoskeleton during oocyte storage and resumption of meiosis.

Another protein affected by IFET-1 depletion was MEI-1/Katanin, a microtubule-severing enzyme important for meiotic spindle function (36). It is not known whether MEI-1 is made before oocyte formation, during oocyte storage, or in response to oocyte activation. We thus used CRISPR to fuse MEI-1 with GFP (GFP::MEI-1) in feminized worms. Confocal microscopy revealed GFP::MEI-1 signal was low in the syncytial gonad and increased 3.5-fold in stored oocytes (Fig 4E and F). The GFP::MEI-1 signal was highest in the most proximal oocytes (i.e., the oldest). Thus, MEI-1 is translated into stored oocytes and accumulates over time. GFP::MEI-1 levels dropped dramatically after IFET-1 depletion (Fig 4E

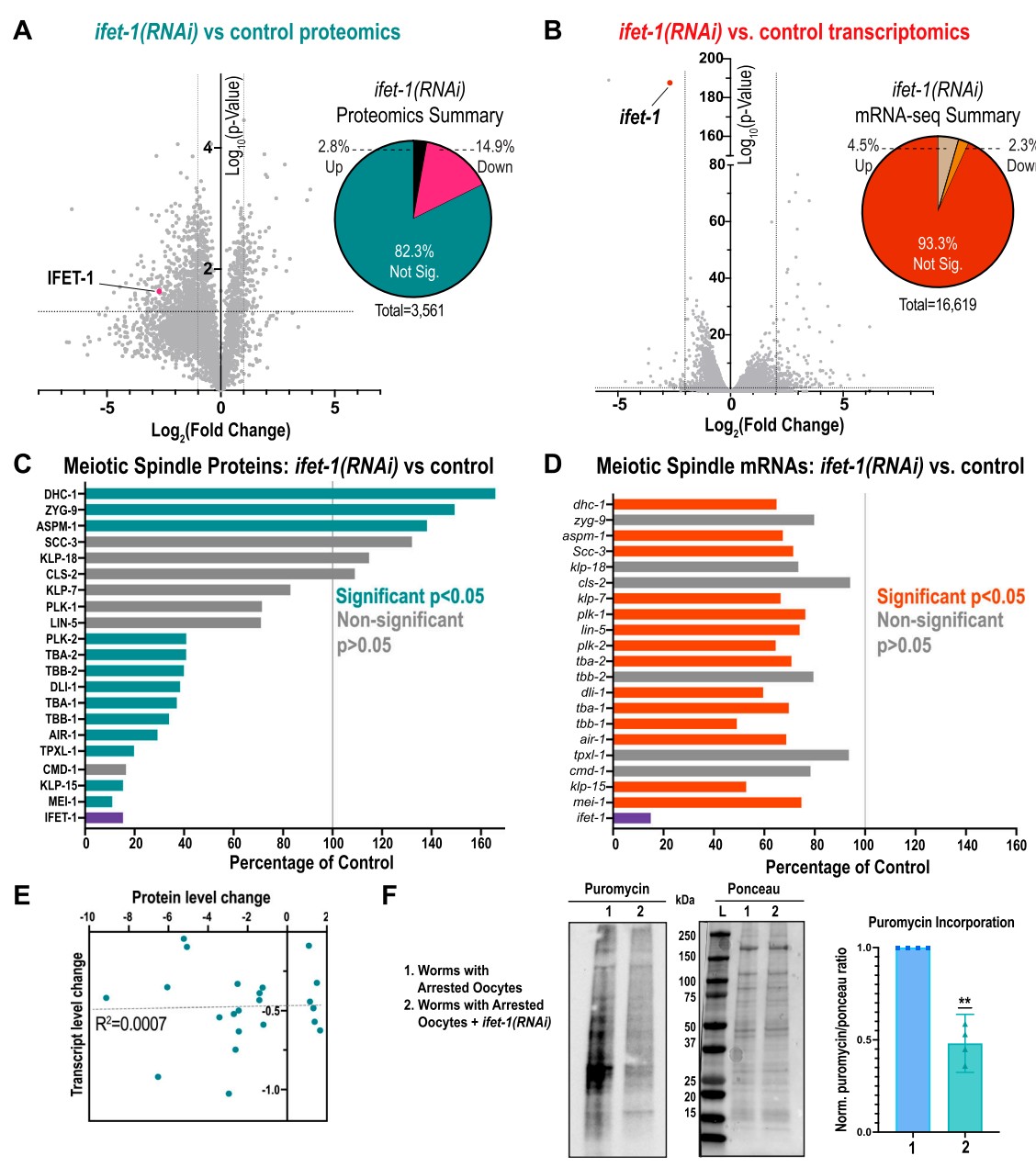

**Figure 3. Proteomic and transcriptomic profiling of oocyte-storing worms lacking IFET-1.**
**(A)** Left, volcano plot of protein abundance in *ifet-1(RNAi)* females versus control females. Right, pie chart indicating protein changes that surpassed significance thresholds ($P < 0.05$, fold change > 2; n = 3 replicates). See Supplemental Data 1 for a list of all proteins. **(B)** Left, volcano plot of mRNA abundance in *ifet-1(RNAi)* females versus control females. Right, pie chart indicating protein changes that surpassed significance thresholds ($P < 0.05$, fold change > 2; n = 3 replicates). See Supplemental Data 2 for a list of all transcripts. **(C)** Analysis of selected proteins related to meiotic spindle formation. Shown is average protein abundance in *ifet-1(RNAi)* worms relative to controls. **(D)** Analysis of selected transcripts related to meiotic spindle formation. Shown is average mRNA abundance in *ifet-1(RNAi)* worms relative to controls. **(C, D, E)** Comparison of changes in protein and mRNA abundance for genes/proteins listed in (C, D). There is no correlation between the data sets ($R^2 = 0.007$; n = 21). **(F)** SUnSET assay testing translation levels by measuring puromycin incorporation into proteins. Lane 1, control female worms with stored oocytes. Lane 2, *ifet-1(RNAi)* female worms with stored oocytes. Left, blot probing for puromycin. Middle, Ponceau stain showing total protein levels per sample. Right, quantification of puromycin signal relative to Ponceau signal and normalized to signal of control worms (n = 4 blots).
Source data are available for this figure.

and F), consistent with our proteomic data. Treatment of adult worms with the proteasome inhibitor bortezomib only weakly restored GFP::MEI-1 levels in the oldest oocytes (Fig 4E and F). Overall, our results suggest that meiotic proteins are produced and maintained in stored oocytes through IFET-1–influenced translation and proteasome-mediated degradation. We conclude that IFET-1 ensures that stored oocytes produce the correct levels of MEI-1/katanin, tubulin, and other proteins needed to organize the microtubule cytoskeleton and, after oocyte activation, to build a functional meiotic spindle.

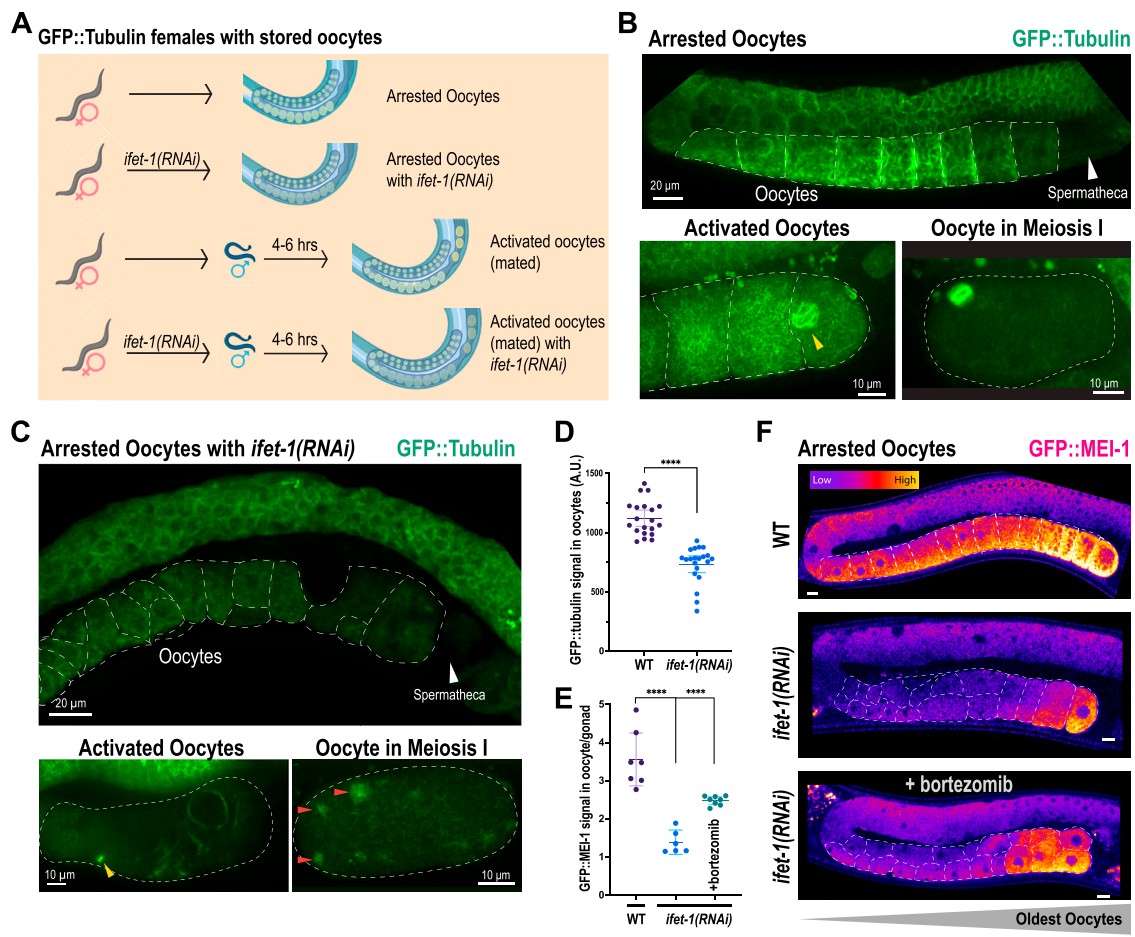

**Figure 4. IFET-1 depletion affects organization of the microtubule cytoskeleton and MEI-1/katanin levels in stored oocytes.**
**(A)** Workflow of experiments in female worms with stored oocytes. Mating induces oocyte activation and meiotic spindle assembly. **(B)** Live-cell confocal images of arrested, activated, and meiosis I oocytes in control female worms expressing GFP::TBB-2 (beta tubulin). Yellow arrowhead, an assembling meiotic spindle. **(B, C)** Same as in (B), except with *ifet-1(RNAi)* female worms. Yellow arrowhead, microtubule asters; red arrowheads, tubulin aggregates. **(D)** Quantification of GFP::TBB-2 levels in arrested oocytes (mean ± 95% C.I.; n = 4–6 worms). **(E)** Quantification of GFP::MEI-1 (katanin) levels in arrested oocytes relative to gonads (mean ± 95% C.I.; n = 6–8 worms). **(F)** Images of GFP::MEI-1 expression in arrested oocytes. Top: arrested oocytes; middle: arrested oocytes with *ifet-1(RNAi)*; bottom: arrested oocytes fed on *ifet-1(RNAi)* and then on bortezomib plates to block the proteasome. Scale bars, 10 μm.

### The homolog of IFET-1 localizes to perinuclear puncta in dormant human oocytes

Our results suggest that IFET-1 plays an oocyte-intrinsic role in maintaining oocyte quality over time in *C. elegans*. We next examined the localization of the human homolog of IFET-1, called EIF4ENIF1. We surgically removed ovarian cortex strips from a reproductively healthy, 20-yr-old patient (Fig 5A; see the Materials and Methods section), and immediately fixed and processed the tissue for immunofluorescence staining. Stored oocytes reside in primordial follicles identified by a single layer of squamous pregranulosa cells (2). Within these oocytes, EIF4ENIF1 localized to the cytoplasm and perinuclear, micron-scale puncta that morphologically resemble condensates (Fig 5B). EIF4ENIF1 puncta did not co-localize with mitochondria, similar to IFET-1 condensates in stored *C. elegans* oocytes (Fig 1). The EIF4ENIF1 puncta were specific to oocytes, as no significant signal was detected in the surrounding tissue (Fig 5C). We conclude that supramolecular assembly of eIF4ET

family proteins into mitochondria-free condensates is a conserved feature of oocyte storage.

## Discussion

Diverse animals store and maintain immature eggs, called oocytes, for long periods of time before ovulating them. Here, we used *C. elegans* females as a short-lived model for oocyte storage and identified the eIF4ET family protein IFET-1 as essential for oocyte formation and long-term maintenance. We report that in stored oocytes, IFET-1 forms condensates and influences proteostasis, ensuring that Katanin, tubulins, and other meiotic spindle proteins are maintained at the correct levels. This program ensures that the oocyte microtubule cytoskeleton is supported during oocyte storage and then properly reorganized into a functional meiotic spindle after oocyte activation.

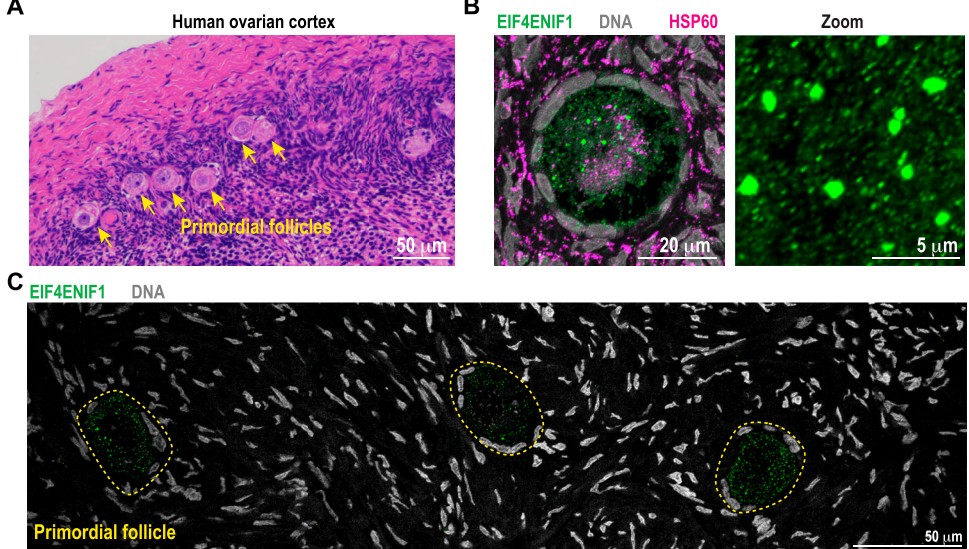

**Figure 5. Localization of the IFET-1 homolog EIF4ENIF1 in human stored oocytes.**
**(A)** H&E staining of an ovarian cortical slice resected from a 20-yr-old patient. Primordial follicles containing stored oocytes are labeled. **(B)** Immunofluorescence staining of mitochondria (HSP60), DNA, and EIF4ENIF1 in a primordial follicle. **(C)** Larger view of the ovary cortex reveals that EIF4ENIF1 only localized to oocytes within the primordial follicle and not surrounding cells.

Oocytes are thought to be largely translationally silent, yet they still produce "housekeeping" proteins, which are presumably needed for survival during long-term storage (37). Our data revealed that IFET-1 is needed for overall translation in oocytes and, more specifically, achieving adequate levels of housekeeping proteins related to microtubule organization and completion of meiosis. IFET-1 co-localized with CAR-1 (Lsm14B) in micron-scale condensates that formed in stored oocytes but not in activated oocytes. Previous work suggested that these condensates serve as super-stoichiometric repressor complexes that allow extreme concentration and silencing of mRNAs (38). If true, one would expect that dissolution of the condensates would cause a massive up-regulation of protein translation. IFET-1 depletion does increase the expression of nonnatural translational reporters (*gfp* fused to different 3′UTRs) (26) and genes required for sperm production (e.g., *fog-1, daz-1, fem-3*) (39). However, polysome profiling revealed that translation is reduced in IFET-1–depleted hermaphrodites (39). Our SUnSET assay also showed reduced translation in *ifet-1(RNAi)* females containing stored oocytes. These seemingly contradictory results are resolved by our unbiased proteomics: 530 proteins are significantly down-regulated, and 100 proteins are significantly up-regulated in *ifet-1(RNAi)* females. Thus, IFET-1 affects protein expression and stability in a protein-specific manner. eIF4ET proteins are known to sequester eIF4E, thus preventing interaction between the 40S ribosomal subunit and the 5′ mRNA cap (19). By repressing translation of capped mRNAs, IFET-1 could, in theory, favor translation of uncapped mRNAs needed for oocyte maintenance (40).

Why loss of IFET-1 would lead to a decrease in global translation is still unclear. There are several caveats to consider. First, for our puromycin and imaging experiments, we performed RNAi knockdown of *ifet-1* before oocyte storage; thus, we currently cannot rule out the possibility that failed translation of some proteins could be an indirect effect of improper oocyte formation. However, our proteomic analysis of *ifet-1(ΔCHD)* females, which produce oocytes normally but cannot effectively store them, revealed decreases in katanin, tubulins, and other cytoskeleton-related proteins. Thus,

translation of these proteins is likely influenced by IFET-1 during oocyte storage. Second, loss of select proteins could be due to increased proteasome-mediated degradation, which we showed is active for maintaining MEI-1 levels in stored oocytes. Third, the primary role of IFET-1 may also change in a cell type–specific manner. Our data, which used proteomics and live-cell imaging of GFP-tagged endogenous MEI-1, clearly indicated that IFET-1 depletion reduces MEI-1 levels in stored oocytes. However, a study using immunofluorescence reported higher levels of MEI-1 in the embryos of *ifet-1* mutant hermaphrodites (41). MEI-1 is normally degraded during the oocyte–embryo transition (42). It is possible that IFET-1 is required for both translation of MEI-1 during oocyte storage and the factors that degrade MEI-1 during the oocyte–embryo transition. Understanding the mechanistic link between IFET-1 and MEI-1 translation will require future investigation. Our data revealed that many other classes of proteins are affected by *ifet-1(RNAi)*; it is thus possible that IFET-1's control of MEI-1 and other microtubule-related protein levels is indirect.

IFET-1 shares homology to human EIF4ENIF1, a protein whose function during mammalian oocyte storage remains poorly defined. Women with heterozygous mutations in *EIF4ENIF1* suffer a form of early-onset infertility called POI (14, 15, 16). Symptoms typically arise in the patient's late 20s and early 30s and include amenorrhea, a rapid depletion of the ovarian reserve of quiescent oocytes, and an elevated risk of miscarriage (43). Before diagnosis, patients with POI can bear healthy children, indicating that POI does not block or perturb oocyte development but rather the long-term maintenance of oocytes in the ovary (13). We showed that mutating conserved regions in IFET-1 specifically impairs oocyte storage but not formation, thus mimicking a key feature of POI. Furthermore, we demonstrated that EIF4ENIF1 localizes to micron-scale condensates in dormant primary oocytes in a patient, similar to our observations of IFET-1 in *C. elegans*. By analogy, it is possible that human oocytes assemble EIF4ENIF1 condensates or smaller cytoplasmic complexes as a quality control mechanism to regulate protein levels during long-term oocyte storage. We hypothesize that

EIF4ENIF1 dysregulation could be an upstream cause of meiotic division errors associated with age-related infertility in women.

## Materials and Methods

### Worm strains and maintenance

All worms were maintained at 16°C on NGM with OP50 until reaching the adult gravid stage and passaged regularly to maintain a steady population of worms. GFP and mMaple knock-ins were generated using CRISPR (44). Worm strains are listed in Table S1.

### Worm synchronization and feminization

All worms with a *fem-1* temperature-sensitive genotype had to be maintained at 16°C to maintain hermaphrodites. When arrested oocytes were needed for an experiment, worms were grown at 16°C until the plate was full of mostly adults and eggs. Plates were washed with 3.5 ml of dH$_2$O and transferred to a Falcon tube. To this, 1 ml of 5% bleach and 0.5 ml of 5M NaOH were added and the Falcon tubes spun for 1–2 min or until all adult worms were dissolved. This embryo-enriched solution was spun on a centrifuge at 1,300 RCF for 1 min, then washed in DI water twice at the same spin settings. The pellet of embryos was resuspended in 1 ml of M9 media and left to rotate at room temperature overnight. The next day, the tubes were collected and contained synchronized L1 hatchlings. These were transferred to NGM plates with OP50 and grown at 25°C for 72 h to fully feminize and mature to adults with oocytes in day 1 arrest. It is at this stage that worms were collected for all subsequent experiments.

### RNAi knockdowns

RNAi was done via feeding. The target sequence for ifet-1 is AACGTGGCTAATCCAATGCTTGCAAGACTTTTCGGTCATAGTGGTGGTGATAA CAACGCTTCATCTTCTGGAGCTGGTGATATAAAAGGAGGAATGCGGTTGGAA GATTTGGAGAAAGGCATGGAGTCAAAGGAACCTTCAAAAGTCAGTCCATTG CAAGATCCATCACAACAAGCACAACTTCTGCAACATCTGCAAAAATTCGCC AAACAGCAAGCAGAGTCTGGTCAGCACATACATCATCATAGACAACCAACAC CACCAAATGGAGGTCCACAGCATCAACAACATTTGCATCACCCTATGGTTCA TCCTGGAATGCAAATTATCGCAGATCCAAGCCTTCTTGCAAGCTTCGCACA- GAATCCAGTGATTTTGAATGCCTATGTCGAGAATCAACTTCAAGAAGCAGTCA ATGCTGCAATTCGTGCAAACAATGGGCAGCAGCTTCCACCACAGCTTCGTATG TTAAGCATTTTTGAATACATTTCATGATTGAATTATTTTTCAGATGAGCAACTT CGTATGGCATCCATGCGTAACAAGGCTTTCCTACAGTCTCAGACTCTCACTTT CGTTAGTCTTCAACAACAGCATCAGAACATCCAGCAGCATCAGCAACAACAA CAGCATCATCAACACAAAGGTCGCACTCCTGCTATGATACCTGCGTCTGTGCA GCGGCAATTGCAGAAGAGCTCCTCAAATGCTGATCAGAAGAAAGAAAAGACG AGTCAGAGCCCGCCAGAGAGCAACCAAGAGACTTCCGACGCTCATAATCAAT CTGATGCCATGAACCAACTGAAGAAATTGCACATGCAGCAGAACTATGCCAA- TATGGTGCAGGCAATGAATTCTGGAGTCGGATGGGCACGCGGAAACGGCGT TGTTAACGGACAACAGCAACATCCACAGCAGCAACTTCCACCAAATGTTCAA ATGCTGATGGCTCAACACCAGCAAGCTCAAATGCAGCACCTCAAG.

Bacteria containing the target sequence plasmids were seeded onto NGM plates containing 1 mM IPTG and 100 µg/ml ampicillin. Worms were bleached and synchronized as described above. After rotating overnight in M9, the L1 larvae were transferred onto prepared RNAi plates and grown at 25°C for 46–72 h (depending on the experiment). Worms were then transferred off the RNAi plate once they reached day 1 of adulthood and allowed to grow on OP50 at 20°C.

### Auxin treatment

Worm synchronization and feminization were conducted as described above. When worms reached day 1 of adulthood, they were transferred to 20°C. Half the adult worms were transferred to NGM plates with OP50 containing 4 mM auxin for 24 h; half were maintained on a normal OP50 plate. After auxin treatment, the worms were transferred back to a regular NGM plate with OP50 for 8 h. Both control and auxin-treated worms were then singled out to mate with three males for ~12–16 h at 16°C. The males used were also synchronized using the above method and grown to adulthood. At the end of the mating period, adult worms were removed from the mating plates and the total amount of embryos on each plate was counted. 2 d later, the number of hatchlings on each plate was counted and % embryonic viability was calculated. Data included were only from worms that laid 34 or less embryos.

For IFET-1 depletion in embryos, embryos were extracted from IFET-1::mScarlet::AID* hermaphrodite worms using two dissection needles in a solution of M9 containing 1 mM auxin. These embryos were then imaged overnight on a Nikon widefield microscope to visualize hatching and IFET-1::mScarlet signal.

### Bortezomib feeding

Agar plates were prepared by diluting bortezomib (#5.04314; MilliporeSigma) to 20 µM in OP-50 and seeding plates as above. Worms were then fed on these plates for 16–24 h to test recovery after RNAi feeding.

### Viability assay

Worms were bleached and synchronized as described above. L1 larvae were grown at 25°C for 72 h to feminize. After 72 h, all adult worms on day 1 of oocyte arrest were transferred to 20°C to reduce heat stress and experiments were conducted on this population. 12–15 d 2 adults were picked and transferred to individual mating plates. fog-2 worms were also synchronized and grown at the same rates and temperatures as the feminized line. Three males per adult female were transferred to each mating plate. Worms were allowed to mate and lay eggs for 10–16 h at 16°C. All adults were then removed, and laid eggs were counted and recorded. 48 h later, all hatchling larvae were counted. Oocyte viability is calculated as the number of hatchlings divided by the number of eggs laid per mother. The average viability per strain is reported for worms that laid ≤34 eggs to maintain a brood size that matches average oocyte storage numbers.

## Western blot

Worms were grown on agar plates until day 2 of adulthood. They were washed off the plate with M9 and allowed to settle at the bottom of the tube. Worms were transferred onto a fresh agar plate with no bacteria. 40 worms of each condition were picked and transferred to 25 µl DI water. 25 µl SDS was added to the samples, and all were boiled at 95° for 5–10 min (or until worms dissolve). Samples were run on SDS–PAGE gel for 30 min. Proteins were transferred onto a nitrocellulose membrane using Trans-Blot Turbo Transfer System (Bio-Rad). Nitrocellulose was blocked in 3% blocking buffer for an hour, and then, primary antibody (1:1,000) was added for overnight incubation at 4°C. Membranes were washed the next morning in TBS-T three times, then incubated with secondary antibody (1:10,000) for an hour. Membranes were washed again with TBS-T three times, and chemiluminescence was expressed using SuperSignal West Femto Maximum Sensitivity Substrate (Thermo Fisher Scientific). Membranes were imaged on ChemiDoc Touch Imaging System (Bio-Rad). Western blots were quantified using FIJI. The mean gray value of any protein bands of interest was measured, and the background mean value of the blot was subtracted. The mean gray value of the control was also measured in the same way (whole lane was measured if it was a Ponceau S stain), and the values of the protein of interest were normalized to the amounts of control or total protein stained.

## Collection, preparation, and staining of human ovarian tissue

Tissue was collected as described previously (45). Briefly, a 20-yr-old female patient with no known pathology or history of infertility was elected for total hysterectomy and bilateral salpingo-oophorectomy for purposes of transgender care. Informed consent was received before the procedure. The use of human ovarian tissue was approved by the Institute Review Board of the University of Texas Southwestern Medical Center (IRB#0801-404/012012-185). The ovarian tissue was placed in 4% formaldehyde directly from the operating room and incubated for 24 h for fixation. The tissue was embedded in paraffin and cut into 5-mm sections using Leica Rotary Microtome. For immunohistochemistry, the mounted paraffin sections were dewaxed and rehydrated in xylene and serial dilutions of ethanol, respectively. After a wash step with PBS, an antigen retrieval step was performed using a basic retrieval reagent (CTS013; R&D) at 90–95°C for 5 min. The slides were cooled completely in PBS before processing.

For immunohistochemistry, the tissue was inserted into a solution containing 10% goat serum, 3% BSA, and 0.03% Triton X-100 (blocking solution) for 1 h. The tissue was then incubated overnight with primary antibody at 4°C. The following primary antibodies were used: directly labeled mouse monoclonal anti-HSP60 (1:500 dilution; 66041-1-Ig; Proteintech) and rabbit polyclonal anti-EIF4ENIF1 (1:500 dilution; ab277638; Abcam). After primary antibody incubation, a series of washes were performed with PBS. The tissue was then incubated with a goat anti-rabbit secondary antibody (1:1,000 dilution, A-11008; Invitrogen) for 1 h before staining for DNA with Vectashield (DAPI) (H-1000-10; VectorLabs). Negative controls were included by incubating with no primary antibody. The directly labeled anti-HSP60 was created using an Alexa Fluor 647 Lightning-Link kit (Abcam) following their instructions.

## Confocal microscopy

Worms were imaged using a Nikon Eclipse Ti2-E microscope with a Yokogawa confocal scanner unit (CSU-W1), piezo Z stage, and an iXon Ultra 888 EMCCD camera (Andor), controlled by Nikon Elements software. Images were collected using a 60X water (NA 1.2) or 100X silicon (NA 1.35) immersion objective, 0.3 µm Z-stacks, 2 × 2 binning, 15% laser power, and 100-ms exposures. Human tissue and worm oocytes were also imaged using a Nikon AX-R confocal microscope with a 40X silicone (NA 1.25) objective.

## Proteomics

Worms were feminized and synchronized as described above and plated on three separate plates at 25°C. When the worms reached day 2 of adulthood, they were washed in M9 media for 15 min to wash off bacteria. For each experimental group, three groups of 30 worms each were collected and transferred to 20 µl of dH₂O in three separate Eppendorf tubes. 20 µl SDS was added to the tubes, and samples were boiled for 5–10 min (or until all worms were dissolved into SDS solution). These samples were run on an SDS–PAGE gel for 3 min to allow all the sample to enter the gel, and then, the gel was stained with a Coomassie dye. The protein band was cut out of the gel.

Gel samples were digested overnight with trypsin (Pierce) after reduction and alkylation with DTT and iodoacetamide (Sigma-Aldrich). After solid-phase extraction cleanup with an Oasis HLB µElution plate (Waters), the resulting peptides were reconstituted in 2% (vol/vol) acetonitrile (ACN) and 0.1% trifluoroacetic acid in water. 1 µg of each sample was injected onto an Orbitrap Fusion Lumos mass spectrometer coupled to an UltiMate 3000 RSLCnano liquid chromatography system (Thermo Fisher Scientific). Samples were injected onto a 75 µm i.d., 75-cm-long EasySpray column (Thermo Fisher Scientific), and eluted with a gradient from 0 to 28% buffer B over 90 min. Buffer A contained 2% (vol/vol) ACN and 0.1% formic acid in water, and buffer B contained 80% (vol/vol) ACN, 10% (vol/vol) trifluoroethanol, and 0.1% formic acid in water. The mass spectrometer operated in positive ion mode with a source voltage of 2.5 kV and an ion transfer tube temperature of 300°C. MS scans were acquired at 120,000 resolution in the Orbitrap, and up to 10 MS/MS spectra were obtained in the Orbitrap for each full spectrum acquired using higher energy collisional dissociation (HCD) for ions with charges 2–7. Dynamic exclusion was set for 25 s after an ion was selected for fragmentation.

Raw MS data files were analyzed using Proteome Discoverer v3.0 (Thermo Fisher Scientific), with peptide identification performed using Sequest HT searching against the *C. elegans* reviewed protein database from UniProt along with the sequence of fluorescent mMaple protein. Fragment and precursor tolerances of 10 ppm and 0.6 D were specified, and three missed cleavages were allowed. Carbamidomethylation of Cys was set as a fixed modification, and oxidation of Met was set as a variable modification. The false discovery rate cutoff was 1% for all peptides.

Data were analyzed in XML format. All protein abundance was normalized by averaging the sum of total proteins in each sample sent and getting a ratio to the average of the control samples sent. The ratios derived were used to adjust the protein abundance values up or down accordingly. From these normalized values, the fold change and *P*-value (two-tailed, unequal variance) were calculated. Anything with a *P*-value below 0.05 was deemed significant. Data were plotted using GraphPad Prism. Identified proteins are listed in Supplemental Data 1. GO-term analysis was conducted using PANGEA (https://www.flyrnai.org/tools/pangea/web/home/6239).

### Transcriptomics

Worms were synchronized, feminized, and grown on two 100-mm plates. When worms reached day 2 of adulthood, they were washed off the plates with M9 and rotated for 20 min to remove all the bacteria. Worms were rinsed twice with M9. Total RNA was extracted from worms with Arcturus PicoPure RNA Isolation Kit (Applied Biosystems). Total RNA was shipped to Novogene. RNA degradation and contamination were monitored on 1% agarose gels. RNA purity was checked using the NanoPhotometer spectrophotometer (IMPLEN). RNA integrity and quantitation were assessed using RNA Nano 6000 Assay Kit of the Bioanalyzer 2100 system (Agilent Technologies).

A total amount of 1 $\mu$g RNA per sample was used as input material for the RNA sample preparations. Sequencing libraries were generated using NEBNext Ultra RNA Library Prep Kit for Illumina (NEB) following the manufacturer's recommendations, and index codes were added to attribute sequences to each sample. Briefly, mRNA was purified from total RNA using poly-T oligo-attached magnetic beads. Fragmentation was carried out using divalent cations under elevated temperature in NEBNext First Strand Synthesis Reaction Buffer (5X). First strand cDNA was synthesized using random hexamer primer and M-MuLV Reverse Transcriptase (RNase H-). Second strand cDNA synthesis was subsequently performed using DNA Polymerase I and RNase H. Remaining overhangs were converted into blunt ends via exonuclease/polymerase activities. After adenylation of 3' ends of DNA fragments, NEBNext Adaptor with a hairpin loop structure was ligated to prepare for hybridization. To select cDNA fragments of preferentially 150~200 bp in length, the library fragments were purified with the AMPure XP system (Beckman Coulter). Then, 3 $\mu$l USER Enzyme (NEB) was used with size-selected, adaptor-ligated cDNA at 37°C for 15 min followed by 5 min at 95°C before PCR. Then, PCR was performed with Phusion High-Fidelity DNA polymerase, universal PCR primers, and index (X) primer. PCR products were purified (AMPure XP system), and library quality was assessed on the Agilent Bioanalyzer 2100 system.

The clustering of the index-coded samples was performed on cBot Cluster Generation System using PE Cluster Kit cBot-HS (Illumina) according to the manufacturer's instructions. After cluster generation, the library preparations were sequenced on an Illumina platform and paired-end reads were generated. Raw data (raw reads) of FASTQ format were firstly processed through fastp. In this step, clean data (clean reads) were obtained by removing reads containing adapter and poly-N sequences and reads with low quality from raw data. At the same time, Q20, Q30, and GC contents

of the clean data were calculated. All the downstream analyses were based on the clean data with high quality.

Reference genome and gene model annotation files were downloaded from the genome website browser (NCBI/UCSC/Ensembl) directly. Paired-end clean reads were aligned to the reference genome using Spliced Transcripts Alignment to a Reference (STAR) software, which is based on a previously undescribed RNA-seq alignment algorithm that uses sequential maximum mappable seed search in uncompressed suffix arrays followed by seed clustering and stitching procedure. STAR exhibits better alignment precision and sensitivity than other RNA-seq aligners for both experimental and simulated data. Differential expression analysis between two conditions/groups (three biological replicates per condition) was performed using the DESeq2 R package. DESeq2 provides statistical routines for determining differential expression in digital gene expression data using a model based on the negative binomial distribution. The resulting *P*-values were adjusted using the Benjamini and Hochberg approach for controlling the false discovery rate. Genes with an adjusted *P*-value < 0.05 found by DESeq2 were assigned as differentially expressed.

Before differential gene expression analysis, for each sequenced library, the read counts were adjusted by trimmed mean of M-values (TMM) through one scaling normalized factor. Differential expression analysis of two conditions was performed using the edgeR R package. The *P*-values were adjusted using the Benjamini and Hochberg methods. Corrected *P*-value of 0.005 and |log2(Fold Change)| of 1 were set as the threshold for significantly differential expression. Identified transcripts are listed in Supplemental Data 2.

### Puromycin assay

Worms were synchronized and grown at 25°C on 100-mm NGM plates until they reached day 2 of adulthood. They were washed off the plates with M9 buffer and rinsed in an Eppendorf tube with S-Basal Medium (100 mM NaCl, 5 mM $K_2HPO_4$, 50 mM $KH_2PO_4$, 5 $\mu$g/ml cholesterol, 10 mM potassium citrate, 3 mM $MgSO_4$, 1% trace metal solution) twice. 250 $\mu$l of S-Basal Medium was added to the live worm pellet along with 50 $\mu$l of 10 mg/ml puromycin and 200 $\mu$l of concentrated OP50 and topped off to 1 ml with S-Basal Medium. The tubes were shaken at 200 RPM for 4 h at room temperature in a Thermomixer (Eppendorf). After 4 h, tubes were collected, and the worms were washed with S-Basal three times. Worms were resuspended in 300 $\mu$l lysis buffer with protease inhibitors (150 mM NaCl, 50 mM Hepes, 1 mM EDTA, 1% Triton X-100, 0.5% SDS) and flash-frozen in liquid nitrogen. The tubes were placed on ice and sonicated five to eight times at 20–30% amplitude for 2 s at a time (until most carcasses were dissolved). They were then spun at 13,000 RPM for 30 min, and 300 $\mu$l of the supernatant was collected. 20% vol/vol of TCA was added to solution, and the protein was allowed to precipitate for 30 min on ice. Tubes were spun at 14,000 RPM for 5 min, and the supernatant was pipetted off. The pellet was washed twice in 90% acetone and then resuspended and boiled in 40 $\mu$l 2X SDS loading buffer. Samples were run on an SDS–PAGE gel, transferred to nitrocellulose, and first stained with Ponceau S stain

and imaged. Membranes were then stained with anti-puromycin antibody (MABE343; MilliporeSigma) and imaged using a ChemiDoc imager (Bio-Rad).

### Immunostaining of *C. elegans*

The tube fixation and collagenase protocol was adapted from protocols in www.wormbook.org. Briefly, worms were fixed in 4% formaldehyde in PBS in an Eppendorf tube. Tubes were flash-frozen in liquid nitrogen and subsequently thawed two to three times. They were shaken at 37°C for 3 h. Fixative was then removed and washed off with PBST. Worms were shaken at 100 rpm in B-mercaptoethanol solution (5% BME, 1% Triton X-100, 120 mM Tris) for 2 h at 37°C. Tubes were spun at 1,000 RCF and washed with PBST up to 10 times if needed, or until fixative is no longer detectable. Samples were transferred to a fresh tube after half the washes. 200 $\mu$l of collagenase solution (100 mM Tris, 1 mM $CaCl_2$, 0.1% Triton X-100, pH 7.4, 2,000 U collagenase) was added, and worms were shaken at 37°C at 250 rpm for 38–48 h (depending on the stock of collagenase). Incubation was stopped when a majority of worms were broken in half. Collagenase was washed off, and worms were blocked in 1 ml blocking buffer (PBS, 0.5% Triton X-100, 1 mM EDTA, 5% BSA, 0.05% sodium azide) at room temperature for 1 h. The blocking buffer was removed, and worms were incubated with primary antibody in blocking buffer (1:1,000) overnight at 4°C. Worms were washed three times for 15-min washes with antibody buffer (blocking buffer but with only 0.1% BSA) and then incubated with secondary antibody in blocking buffer (1:10,000) for 3 h at room temperature. Three more 15-min washes were done and worms transferred onto a slide with an agar pad or wire spacing. Mounting medium was added, and slides were imaged on a spinning disk confocal microscopy. Rabbit polyclonal anti-IFET-1 and anti-CAR-1 antibodies were made by GenScript.

### Statistical analyses

Statistical analyses were performed using GraphPad Prism. The indicated tests, the n values, and significant values are displayed in the figure legends or on the figures themselves. Significance was reported as follows: *$P < 0.05$, **$P < 0.005$, ***$P < 0.0005$, ****$P < 0.00005$.

## Data Availability

The Supplemental Material contains processed proteomics data (Supplemental Data 1) and identified transcripts and their abundance from RNA-seq (Supplemental Data 2). Oocyte viability assays and GO-term analyses are available in the Supplemental Material (Supplemental Data 3 and Supplemental Data 4). Other data are available upon request.

## Supplementary Information

## Acknowledgements

We thank the UTSW Proteomics Core Facility for performing mass spectrometry, the UTSW Histology Core for sectioning human tissue, and Abby Dernberg (UC Berkeley) for the mMaple plasmid. This project was funded by a Pew Biomedical Scholars award, a Welch Catalyst award (V-I-0004-20230731), a Human Frontier Science Program fellowship to K Yaguchi (LT0064/2022-L), and the Endowed Scholars program at UT Southwestern (to JB Woodruff). R Amin and O Bukulmez are supported by an Obstetrics and Gynecology departmental fund.

## Author Contributions

P Bhatia: data curation, formal analysis, investigation, visualization, methodology, and writing—review and editing.
J Tafur: investigation and methodology.
R Amin: investigation and methodology.
NE Familiari: investigation and methodology.
K Yaguchi: methodology.
VM Tran: methodology.
A Bond: methodology.
O Bukulmez: conceptualization, resources, supervision, funding acquisition, project administration, and writing—review and editing.
JB Woodruff: conceptualization, resources, data curation, supervision, funding acquisition, visualization, methodology, project administration, and writing—original draft, review, and editing.

## Conflict of Interest Statement

The authors declare that they have no conflict of interest.

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
