## [Reviewer comments · Life Science Alliance]

Life Science Alliance

Condensate-forming eIF4ET ensures adequate levels of meiotic proteins to support oocyte storage

Priyankaa Bhatia, Judith Tafur, Ruchi Amin, Nicole E. Familiari, Kan Yaguchi, Vanna Tran, Alec Bond, Orhan Bukulmez and Jeffrey B Woodruff
DOI: <https://doi.org/10.26508/lsa.202503387>

Corresponding author(s): Dr. Jeffrey B Woodruff (The University of Texas Southwestern Medical Center)

Review Timeline:	Submission Date:	2025-05-14
	Editorial Decision:	2025-05-16
	Revision Received:	2025-05-19
	Editorial Decision:	2025-05-20
	Revision Received:	2025-05-20
	Accepted:	2025-05-21

Scientific Editor: Tim Fessenden

Transaction Report:

Please note that the manuscript was previously reviewed at another journal and the reports were taken into account in the decision-making process at *Life Science Alliance*.

Reviews

Reviewer #1 Review

Comments to the Authors (Required):

The authors have added new experiments that make a strong case for a requirement for IFET-1 in fully-grown oocytes. A key new experiment uses an auxin degnon to acutely remove IFET-1 in oocytes after they have completed development and are mostly fully grown - the authors show that auxin-induced degradation of IFET-1 at this stage dramatically reduces embryonic viability. In contrast depletion of IFET-1 from embryos (as young as 1-cell stage) did not affect viability. These experiments demonstrate a requirement for IFET-1 in mature oocytes.

However, what IFET-1, a presumed translational repressor, does in mature oocytes to guarantee their viability remains unclear. The authors use proteomics (and direct observation of the katanin MEI-1) to show that IFET-1 depletion by RNAi appears to cause a loss of proteins that normally accumulate in mature oocytes, including proteins like MEI-1 that regulate microtubule function. The problem is that these experiments use RNAi to remove IFET-1, instead of the auxin degnon. RNAi removes IFET-1 from BOTH developing and fully grown oocytes complicating experiment interpretation. Prior studies have shown that IFET-1(RNAi) interferes with oogenesis and causes germ cells to upregulate male transcripts (Huggins et al., 2020). In fact, consistent with these prior studies, in Figure 4F, it appears that ifet-1(RNAi) females contain fewer fully grown oocytes compared to wild-type. The apparent loss of MEI-1, and other proteins identified in the proteomic experiments is likely, therefore, to be an indirect effect of the reduced number of grown oocytes in ifet-1(RNAi) germlines. This interpretation provides an explanation for why loss of a translational repressor, leads to a reduction, rather than a gain, of many proteins.

The authors can address this problem by repeating key experiments using their new IFET-1::mScarlet::AID allele to directly examine the effect of acute depletion of IFET-1 in mature oocytes using auxin. Fig S1D shows that, unlike with RNAi, IFET-1::mScarlet::AID females treated with auxin maintain fully grown arrested oocytes and are therefore a perfect experimental system to examine the role of IFET-1 in arrested oocytes. The authors could use those conditions to examine whether IFET-1 is required for MEI-1 accumulation, for example. Ideally, they would also repeat the proteomic experiments under these conditions as well. At a minimum, the authors need to demonstrate that the effects they observe under RNAi conditions are NOT due to a reduction in the number of mature oocytes.

Below are specific suggestions.

1. The authors should show the outline of oocytes in every figures so the reader can assess whether arrested oocytes are present. Fig. 4F appears to show that IFET-1(RNAi) gonads contain fewer mature oocytes compared to WT, which would explain the apparent lack of MEI-1. The authors should avoid using RNAi depletion unless they can demonstrate that arrested oocytes are present and can be examined under these conditions.
2. The authors make a good case that IFET-1 is required in fully grown oocytes for their viability using auxin-depletion of IFET-1 in females containing arrested, fully mature oocytes. Whether IFET-1 is required SPECIFICALLY for long-term storage of oocytes is less clear since the degnon-experiments were only done in aged oocytes. What happens if IFET-1 is depleted from "fresh" oocytes (oocytes in hermaphrodites)? One possibility is that IFET-1 is required for the production of MEI-1 and other microtubule proteins during the last stages of oogenesis (oocyte growth/maturation), AS WELL AS continued production during oocyte storage. The authors should address this possibility.
3. The most convincing data that supports a role for IFET-1 SPECIFICALLY during oocyte storage is the demonstration that mutants with reduced IFET-1 levels in arrested oocytes do not affect the quality of "fresh" oocytes (oocytes in hermaphrodites) but compromise the quality of "stored" oocytes (oocytes in females). It would be interesting to examine MEI-1 levels in these mutants to confirm that these IFET-1 mutants specifically compromise MEI-1 production under prolonged storage conditions.
4. The authors should clarify in the text that the puromycin assays were done on whole worms and do not necessarily report therefore on translation specifically in oocytes. I suggest they show in the SAME main figure the comparisons between non-arrested vs. arrested oocytes (aka. hermaphdites vs females), and WT arrested vs ifet-1(RNAi) arrested (WT females versus ifet-1(RNAi)). Currently these data are split between Fig. 3F and a supplementary figure S3E. Again it is important in these experiments to demonstrate that the same number of arrested oocytes are present in all conditions, including ifet-1(RNAi).

5. Auxin is not permeable to the eggshell of embryos <https://pmc.ncbi.nlm.nih.gov/articles/PMC9208642/>. How were the experiments shown in Fig. S1F done????

Reviewer #2 Review

Comments to the Authors (Required):

I thank the authors for their response and for incorporating previously omitted references into the revised manuscript. However, it is noteworthy that both reviewers raised concerns regarding the omission of key literature. The authors' admission that these references were initially overlooked-and their subsequent inclusion-substantially alters the framing of the manuscript. Specifically, it reveals that much of what is presented here has already been described, particularly with respect to EIF4ENIF1's interaction with CAR-1/LSM14B-like proteins, the fact that it has a role in translational regulation, and the formation of RNP condensates during oocyte development.

While the authors differentiate their study through a focus on the oocyte storage phase and the use of conditional depletion (e.g., IFET-1-AID), these distinctions, while valid, could be seen as incremental rather than conceptually novel. The core mechanistic themes remain broadly aligned with prior findings.

In conclusion, the manuscript is now more accurately positioned within the existing body of work, and the authors have formally addressed my initial concerns. However, the revised version also clarifies that the novelty of the study lies more in its application of established concepts to a specific developmental window than in the discovery of fundamentally new biology.

Reviewer #1 Review

Comments to the Authors (Required):

This paper reports on the role of IFET-1, a 4E-T homolog in *C. elegans*. Previous studies showed that IFET-1 functions with its binding partner, the cap binding protein IFE-3, as a broad scale regulator of mRNA translation during oogenesis (Sengupta et al., 2013). Among other targets, IFET-1 represses the translation of spermatogenic factors (Huggins et al., 2020 <https://pubmed.ncbi.nlm.nih.gov/32079657/>) as well as the katanin MEI-1 (microtubule-severing protein), which is required in early embryos to shape meiotic spindles (Li et al., 2009 <https://pubmed.ncbi.nlm.nih.gov/19786575/>). Animals that lack IFET-1 show many morphological germline defects, including abnormal oocytes and dead embryos.

In this new study, the authors suggest that IFET-1 additionally plays a role in maintaining the quality of fully grown oocytes during storage. The authors use RNAi to partially deplete IFET-1 and claim that a 48hr treatment gives rise to "normal oocytes" that are specifically deficient in the pro-storage function of IFET-1. The authors provide no direct evidence for this claim. For example, they do not show that the RNAi treatment specifically depletes IFET-1 from oocytes AFTER these have formed, so the possibility that these oocytes are born defective due to lack of IFET-1 during development remains. The assay that the authors use to examine the viability of arrested oocytes also does not make sense: *C. elegans* females only maintain ~8 arrested oocytes per gonadal arm, which are laid within the first two hours after mating. The authors collected progeny of mated females for 16-20 hours, which means that the majority of the progeny they examined were derived, not from arrested oocytes, but from continuously growing oocytes derived from immature germ cells. The authors also claim that IFET-1-depleted oocytes have reduced levels of meiotic proteins including MEI-1 - they fail to cite Li et al., 2009, who already identified MEI-1 as miss-regulated in IFET-1 (aka *spn-2*) mutants.

The most interesting finding is that IFET-1 mutants lacking specific domains apparently exhibit low oocyte viability in aged but not young females (Fig.4B). This is an interesting finding, but without full information regarding brood sizes etc.. (see below), this result, while intriguing, does not appear sufficiently developed.

In summary, this manuscript does not adequately cite the literature and fails to exclude the possibility that the phenotypes they observe are due to the already documented role of IFET-1 in oocyte development.

Specific comments:

- How does the 48hr RNAi treatment deplete IFET-1 levels in oocytes, without affecting IFET-1 levels during oogenesis (even though the RNAi treatment is started at the first larval stage)? Also, how does the RNAi treatment maintain IFET-1 OFF for several days even though the RNAi treatment was stopped after 2 days?
- Please provide the number of embryos scored in all assays, not just the percent viability. Use scatter plots to represent the data, including independent replicates. This is particularly important for Fig. 2B which only shows % viability and no actual embryo counts. Were brood sizes similar between WT and the IFET-1 mutants?
- Fig. 2: The mutant proteins accumulate at lower levels compared to wild-type and therefore their lower levels could have affected oocyte development before storage. Could the difference in viability between young and older females be accounted by differences in IFET-1 protein accumulation?
- What is the evidence that oogenesis occurs normally in IFET-1 depleted and IFET-1 mutant animals? For example, both the oocytes and pachytene region shown in figure 4C appear abnormal. What is the evidence that IFET-1 is required specifically for storage?
- Please cite the full literature including:
 - o Li et al., 2009 <https://pubmed.ncbi.nlm.nih.gov/19786575/> for evidence that IFET-1 regulates MEI-1
 - o Huggins et al., 2020 <https://pubmed.ncbi.nlm.nih.gov/32079657/> for evidence that IFET1 suppress sperm-promoting genes during oogenesis.
 - o Noble et al., 2008 <https://pubmed.ncbi.nlm.nih.gov/18695046/> for "grP bodies", which contain CAR-1 and other RNA binding proteins found in P bodies. The authors report that IFET-1 localizes to CAR-1 granules but do not mention this work.

Line 154, "worm" should be plural.

Line 211, "mutant" should be RNAi.

Reviewer #2 Review

Comments to the Authors (Required):

Bhatia et al. investigated the role of iFET-1 in *C.elegans* oocyte formation and storage. They found that iFET-1 deletion leads to defects in both oocyte formation and storage. iFET-1 protein seems to function at least in part by increasing translation in stored *C.elegans* oocytes.

The authors' data on the characterization of iFET-1 function are robust, and the manuscript is well-written with mostly well-executed experiments. However, I have several major concerns, which are outlined below.

1- The authors omit a significant body of literature when claiming that the role of EIF4ENIF1 in oocytes is not defined. For example, they fail to cite a seminal study conducted on mouse oocytes with very similar findings to theirs (Cheng et al., 2022, *Science*; PMID: 36264786). This study demonstrated that EIF4ENIF1 forms condensates with LSM14B (CAR-1 in *C.elegans*) also in mouse oocytes, and these condensates play a crucial role in protein translation by repressing mRNA translation. In effect, Cheng et al. reached conclusions that are directly parallel to those of the current work, yet this important reference is not mentioned at all.

There are several others studies that link EIF4ENIF1 function to oocyte defects in the literature such as PMID: 38088064 (mouse), PMID: 26015597 (*Xenopus*) and others...

2- The authors seemed to have cherry-picked meiotic proteins from their proteomics experiment in Figure 3A to conclude that IFET-1 regulates meiotic proteome. What would an unsupervised analysis highlight? From the volcano plot, it is very clear that the depletion of iFET-1 affects hundreds of proteins.

Meiotic spindle is very sensitive to changes in cellular homeostasis, and its defects could be a downstream effect of more pronounced changes in oocytes. Therefore, the authors' major claim (and also title) is not supported by their current data.

3- SunSET assay for active and arrested oocytes should be run on the same gel for proper comparison.

4- At line 53 in introduction, the authors seem to confuse early oocyte quiescence with transcriptional silencing observed during meiotic maturation. Luong et al., studies only the short time period (18 hours) around meiotic maturation, not the longer term oocyte storage as in Rodriguez-Nuevo et al.

Other concerns

1- IFET1-mMaple colocalization with endogenous CAR-1 seems very poor although the text claims colocalization.

We have carefully considered the comments and now submit a revised version of our manuscript. While extensive details exist about how oocytes mature and transition to embryos, the life support mechanisms allowing arrested oocytes to survive long-term storage are poorly understood. Our study reveals that the conserved 4E transporter protein IFET-1 supports translation and accumulation of meiotic spindle and microtubule-related proteins during oocyte storage in *C. elegans*. We feel that our study makes several important advances in the field:

1. We provide two experiments that demonstrate the necessity of IFET-1 during oocyte storage: 1) separation of function mutants and 2) a new conditional knock-down. These experiments have not been performed before in any species.
2. We report that stored oocytes are actively translating cytoskeletal proteins. Previous assumptions were that dormant oocytes are translationally silent. Thus, our work adds to a newly emerging body of work (Pan et al., 2005; Rodriguez-Nuevo et al., 2022) showing that “dormant” oocytes are actually metabolically active.
3. We report a novel role for 4E-T in promoting translation in stored oocytes. This is contrast to the reported repressive role of 4E-T proteins during oocyte maturation and in somatic cells. Thus, our work demonstrates that the activities of RNPs are context-dependent.

Reviewer Concerns:

The main concerns boiled down to two points: 1) insufficient evidence demonstrating the importance of IFET-1 during oocyte storage, and 2) failure to cite key literature documenting the role of IFET-1/4E-T homologs in maturing/growing oocytes.

We have addressed all concerns from the reviewers. We have included several major changes including:

1. **Conditional knockdown of IFET-1 specifically during oocyte storage (Figure 1C).** We used CRISPR to create female worms expressing degron-tagged IFET-1 (IFET-1::mScarlet::AID*). By incubating adult female worms (which contain pre-formed, healthy stored oocytes) on auxin-containing plates, we see near complete depletion of IFET-1 within the stored oocytes. We then take the worms off auxin for 8 hr (to metabolize and clear out the auxin), let them mate with males, then measure viability of the offspring. Aged oocytes depleted of IFET-1 were almost entirely inviable (avg = 9% viable). However, embryos treated with auxin were viable, indicating that IFET-1 is required in stored oocytes but not during embryogenesis. This finding supports our separation-of-function experiments (Figure 2) in revealing that IFET-1 plays a role specifically during

oocyte storage.

2. **Unsupervised analysis of our proteomic data (Figure S3).** GO-term analyses show that multiple processes are affected in ifet-1(RNAi) worms. Importantly, the top affected category contains meiotic spindle and microtubule-related proteins.
3. **Proper citations of previous studies that documented IFET-1/4E-T during oocyte maturation.**

Sincerely,
Jeff Woodruff

References

- Pan, H., J. O'Brien M, K. Wigglesworth, J.J. Eppig, and R.M. Schultz. 2005. Transcript profiling during mouse oocyte development and the effect of gonadotropin priming and development in vitro. *Dev Biol.* 286:493-506.
- Rodriguez-Nuevo, A., A. Torres-Sanchez, J.M. Duran, C. De Guirior, M.A. Martinez-Zamora, and E. Boke. 2022. Oocytes maintain ROS-free mitochondrial metabolism by suppressing complex I. *Nature.* 607:756-761.

Specific Responses to Reviewers:

Reviewer #1 (Comments to the Authors (Required)):

This paper reports on the role of IFET-1, a 4E-T homolog in *C. elegans*. Previous studies showed that IFET-1 functions with its binding partner, the cap binding protein IFE-3, as a broad scale regulator of mRNA translation during oogenesis (Sengupta et al., 2013). Among other targets, IFET-1 represses the translation of spermatogenic factors (Huggins et al., 2020 <https://pubmed.ncbi.nlm.nih.gov/32079657/>) as well as the katanin MEI-1 (microtubule-severing protein), which is required in early embryos to shape meiotic spindles (Li et al., 2009 <https://pubmed.ncbi.nlm.nih.gov/19786575/>). Animals that lack IFET-1 show many morphological germline defects, including abnormal oocytes and dead embryos.

In this new study, the authors suggest that IFET-1 additionally plays a role in maintaining the quality of fully grown oocytes during storage. The authors use RNAi to partially deplete IFET-1 and claim that a 48hr treatment gives rise to "normal oocytes" that are specifically deficient in the pro-storage function of IFET-1. The authors provide no direct evidence for this claim. For example, they do not show that the RNAi treatment specifically depletes IFET-1 from oocytes AFTER these have formed, so the possibility that these oocytes are born defective due to lack of IFET-1 during development remains. The assay that the authors use to examine the viability of arrested oocytes also does not make sense: *C. elegans* females only maintain ~8 arrested oocytes per gonadal arm, which are laid within the first two hours after mating. The authors collected progeny of mated females for 16-20 hours, which means that the majority of the progeny they examined were derived, not from arrested oocytes, but from continuously growing oocytes derived from immature germ cells.

We have addressed this concern via acute knockdown of IFET-1 in oocytes after they have already formed (Figure 1C-F).

In the revised version, we include a new experiment using a conditional mutant of IFET-1 (IFET-1::mScarlet::AID*) that can be depleted with auxin addition after oocytes are formed. The results show 1) oocytes look normal prior to auxin treatment (as they should), 2) auxin treatment almost completely depletes IFET-1 within the stored oocytes, and 3) the auxin treated oocytes fail to make viable embryos. Furthermore, we demonstrate that specifically depleting IFET-1 in embryos does not affect their viability (Figure S1F,G). We believe this demonstrates that IFET-1 is essential for the maintenance of oocytes during their storage and not for embryogenesis.

We still believe in the value of our original viability assay (Figure 1A,B). In our Day 2 adult females, we counted on average 34 oocytes total per mother (17 per gonad arm). This is seen in Figure 1D or Figure S1D. The reviewer's count may refer to hermaphrodites, or perhaps younger females that have not had the time to make as many oocytes. In the revised draft, we now specify our conditions (time, temperature) that led to our worms producing and storing 34 oocytes. We also adjusted the egg laying time to ensure that fewer than 34 eggs were laid. Thus, the data shown in our viability assays most likely reports on stored oocytes and not oocytes that are formed after the stored oocytes have been ovulated.

In addition, we provide a new proteomic analysis of the *ifet-1*(Δ CHD) mutant, which can produce oocytes normally but not store them. This mutant has lower levels of beta

tubulin, MEI-1, and other meiotic proteins, similar to, but less severe than, *ifet-1(RNAi)* worms (Figure S3D). This result further supports our conclusion that IFET-1 is required for full accumulation of meiotic cytoskeletal proteins in stored oocytes.

The authors also claim that IFET-1-depleted oocytes have reduced levels of meiotic proteins including MEI-1 - they fail to cite Li et al., 2009, who already identified MEI-1 as miss-regulated in IFET-1 (aka *spn-2*) mutants.

We apologize for not including this citation. Our revised version mentions this study in the discussion.

The most interesting finding is that IFET-1 mutants lacking specific domains apparently exhibit low oocyte viability in aged but not young females (Fig.4B). This is an interesting finding, but without full information regarding brood sizes etc.. (see below), this result, while intriguing, does not appear sufficiently developed.

We have provided additional characterization of these mutants (see specific comments below) and provided all raw data, including brood sizes, in Extended Data Set 3.

In summary, this manuscript does not adequately cite the literature and fails to exclude the possibility that the phenotypes they observe are due to the already documented role of IFET-1 in oocyte development.

With our new experiments, we show that IFET-1 is required during the storage phase and that it promotes translation of meiotic cytoskeletal proteins. We believe that IFET-1 thus constitutes a translational program to maintain the cytoskeleton during storage and quickly rearrange it upon oocyte activation. These insights have not been reported previously and go beyond the studies connecting IFET-1 to oocyte formation or maturation following activation.

Specific comments:

- How does the 48hr RNAi treatment deplete IFET-1 levels in oocytes, without affecting IFET-1 levels during oogenesis (even though the RNAi treatment is started at the first larval stage)?

The RNAi treatment does partly deplete IFET-1 levels during oogenesis, thus resulting in the main drawback of our study. We could not perform RNAi starting with adults containing oocytes, as the oocytes already contain a maternally supplied pool of IFET-1 protein. To resolve this issue, we now perform conditional knockdown of IFET-1 after oocytes are formed (Figure 1C-F; IFET-1-degron; see above for an in-depth description).

Also, how does the RNAi treatment maintain IFET-1 OFF for several days even though the RNAi treatment was stopped after 2 days?

We show that IFET-1 is not detectable even after worms are removed from *ifet-1*(RNAi) feeding plates (Figure S1C). This is likely because the maternal mRNA encoding IFET-1 has been depleted and transcription/translation of *ifet-1* is silenced in the stored oocytes.

- Please provide the number of embryos scored in all assays, not just the percent viability. Use scatter plots to represent the data, including independent replicates. This is particularly important for Fig. 2B which only shows % viability and no actual embryo counts. Were brood sizes similar between WT and the IFET-1 mutants?

We have provided all raw data for the viability assays in Extended Data Set 3. We thought that including all information, as suggested by the reviewer, would make the main figures too busy and difficult to interpret.

- Fig. 2: The mutant proteins accumulate at lower levels compared to wild-type and therefore their lower levels could have affected oocyte development before storage. Could the difference in viability between young and older females be accounted by differences in IFET-1 protein accumulation?

We tested the reviewer's hypothesis by performing western blots (Figure S2A,B). Please keep in mind that we compared age-matched hermaphrodites (oocytes always active) vs. females (stored oocytes for 2 days). Hermaphrodite mothers with active oocytes and female mothers with arrested oocytes express the same amounts of mutant IFET-1 proteins. However, the viability of their oocytes are remarkably different. This suggests that the difference in viability is not due to IFET-1 protein levels per se, but rather a distinct need for fully functional IFET-1 during oocyte storage that is not required for oocyte formation or maturation.

- What is the evidence that oogenesis occurs normally in IFET-1 depleted and IFET-1 mutant animals? For example, both the oocytes and pachytene region shown in figure 4C appear abnormal. What is the evidence that IFET-1 is required specifically for storage?

We agree with the review that our original study did not adequately demonstrate a role for IFET-1 specifically during oocyte storage. Thus, in the revised version, we include a new experiment using a conditional mutant of IFET-1 (IFET-1::mScarlet::AID*) that can be depleted after oocytes are formed (Figure 1C-F). See above for a more detailed explanation.

- Please cite the full literature including:

- o Li et al., 2009 <https://pubmed.ncbi.nlm.nih.gov/19786575/> for evidence that IFET-1 regulates MEI-1
- o Huggins et al., 2020 <https://pubmed.ncbi.nlm.nih.gov/32079657/> for evidence that IFET1 suppress sperm-promoting genes during oogenesis.
- o Noble et al., 2008 <https://pubmed.ncbi.nlm.nih.gov/18695046/> for "grP bodies", which contain CAR-1 and other RNA binding proteins found in P bodies. The authors report

that IFET-1 localizes to CAR-1 granules but do not mention this work.

We have cited these references and apologize for overlooking them in the previous draft.

Line 154, "worm" should be plural.

Line 211, "mutant" should be RNAi.

We have corrected these.

Reviewer #2 (Comments to the Authors (Required)):

Bhatia et al. investigated the role of iFET-1 in *C.elegans* oocyte formation and storage. They found that iFET-1 deletion leads to defects in both oocyte formation and storage. iFET-1 protein seems to function at least in part by increasing translation in stored *C.elegans* oocytes.

The authors' data on the characterization of iFET-1 function are robust, and the manuscript is well-written with mostly well-executed experiments. However, I have several major concerns, which are outlined below.

1- The authors omit a significant body of literature when claiming that the role of EIF4ENIF1 in oocytes is not defined. For example, they fail to cite a seminal study conducted on mouse oocytes with very similar findings to theirs (Cheng et al., 2022, *Science*; PMID: 36264786). This study demonstrated that EIF4ENIF1 forms condensates with LSM14B (CAR-1 in *C.elegans*) also in mouse oocytes, and these condensates play a crucial role in protein translation by repressing mRNA translation. In effect, Cheng et al. reached conclusions that are directly parallel to those of the current work, yet this important reference is not mentioned at all.

We have revised our manuscript to emphasize that much is known about EIF4ENIF1 during oocyte maturation. EIF4ENIF1 does co-localize with CAR-1-like proteins in the MARDO, which is a mitochondria-associated RNP in maturing mammalian oocytes. We now cite this reference. Please see the added sections in the introduction (highlighted in yellow).

We would like to point out several things that differentiate our findings from that of Cheng et al. 2022. First, we present a role for IFET-1 specifically during oocyte storage. In Figure 2, we showed that several mutants of IFET-1 affect oocyte storage without affecting oocyte formation and maturation (e.g., non-stored oocytes were perfectly healthy, whereas two-day stored oocytes were largely inviable). In our revised manuscript, we perform a conditional knockdown of IFET-1 (Figure 1; IFET-1-AID). We show that pre-formed, healthy oocytes require a supply of IFET-1 to produce healthy offspring. To our knowledge, conditional knockdown of EIF4ENIF1-related proteins has not been previously performed. Second, our proteomics and SUNSET assay reveal that IFET-1 promotes translation in stored oocytes and ensures adequate levels of tubulin,

katanin, kinesins and other proteins needed to support the microtubule cytoskeleton. Thus, the IFET-1 condensates in stored oocytes likely play a different role than the MARDO does in maturing oocytes (where it represses translation). Third, we show that IFET-1 condensates do not contain or surround mitochondria, thus making them a different kind of structure than the MARDO or Balbian Body.

There are several others studies that link EIF4ENIF1 function to oocyte defects in the literature such as PMID: 38088064 (mouse), PMID: 26015597 (Xenopus) and others...

We apologize for not including these references before. We include these references in the revised manuscript regarding EIF4ENIF's role in oocyte formation and maturation following release from storage. Please note that these studies did not assess the role of EIF4ENIF1 during oocyte storage (primordial to GV stage), which we now emphasize in our revised manuscript.

2- The authors seemed to have cherry-picked meiotic proteins from their proteomics experiment in Figure 3A to conclude that IFET-1 regulates meiotic proteome. What would an unsupervised analysis highlight? From the volcano plot, it is very clear that the depletion of iFET-1 affects hundreds of proteins.

We now include a broader, unsupervised analysis of our proteomics in revised Figure S3C. When grouping significant hits based on molecular function, the top group relates to Meiosis and embryo development. This category includes the proteins we focused on in Figure 3. We acknowledge that other protein classes are affected, and that these could contribute to oocyte quality.

Meiotic spindle is very sensitive to changes in cellular homeostasis, and its defects could be a downstream effect of more pronounced changes in oocytes. Therefore, the authors' major claim (and also title) is not supported by their current data.

We chose to follow up on the microtubule-related proteins because they are the most obvious connection to oocyte quality. Chromosomal abnormalities, likely caused by defective meiotic spindle function, are well-documented to occur in aging and low-quality oocytes. Our results indicate that IFET-1 is an upstream factor that helps

oocytes achieve correct levels of these meiotic proteins. Of course, there could be many factors in between IFET-1's role in translation and levels of Katanin, tubulins, etc. We now add a caveat in the revised discussion (see highlighted text). We have also revised the title to be more accurate.

3- SunSET assay for active and arrested oocytes should be run on the same gel for proper comparison.

The SUNsET assay for active and arrested oocytes is shown on the same gel in Figure S3C.

4- At line 53 in introduction, the authors seem to confuse early oocyte quiescence with transcriptional silencing observed during meiotic maturation. Luong et al., studies only the short time period (18 hours) around meiotic maturation, not the longer term oocyte storage as in Rodriguez-Nuevo et al.

We have fixed this error and replaced the Luong reference with Moore et al., 1974, who showed that transcription is low in mouse primordial oocytes.

Other concerns

1- IFET1-mMaple colocalization with endogenous CAR-1 seems very poor although the text claims colocalization.

We acknowledge that the co-localization was hard to see, likely due to the color choice and size of the images. In our revised draft, we now change the coloration to green/magenta and include zoomed in images to make the co-localization easier to see (Figure 1H).

May 16, 2025

Re: Life Science Alliance manuscript #LSA-2025-03387-T

Dr. Jeffrey B Woodruff
UT Southwestern
Cell Biology
6000 Harry Hines Blvd
Dallas, TX 75235

Dear Dr. Woodruff,

Thank you for transferring your manuscript entitled "Condensate-forming eIF4ET ensures adequate levels of meiotic proteins to support oocyte storage" to Life Science Alliance. As indicated in our offer, we invite you to submit a revised manuscript addressing the following reviewer points which we will assess without further review:

- Amend the text to acknowledge the main limitation raised by Reviewer 1, that the effects observed under RNAi conditions may reflect a reduction in the number of mature oocytes.
- Modify figures according to Point 1 by Reviewer 1.
- Modify the text according to Point 4 by Reviewer 1.

While you are revising your manuscript, please also attend to the below editorial points to help expedite the publication of your manuscript. Please direct any editorial questions to the journal office. When submitting the revision, please include a letter addressing the reviewers' comments noted above.

Thank you for this interesting contribution to Life Science Alliance. We are looking forward to receiving your revised manuscript.

Sincerely,

- An editable version of the final text (.DOC or .DOCX) is needed for copyediting (no PDFs).
- High-resolution figure, supplementary figure and video files uploaded as individual files: See our detailed guidelines for preparing your production-ready images, <https://www.life-science-alliance.org/authors>
- Summary blurb (enter in submission system): A short text summarizing in a single sentence the study (max. 200 characters including spaces). This text is used in conjunction with the titles of papers, hence should be informative and complementary to the title and running title. It should describe the context and significance of the findings for a general readership; it should be written in the present tense and refer to the work in the third person. Author names should not be mentioned.
- By submitting a revision, you attest that you are aware of our payment policies found here: <https://www.life-science-alliance.org/copyright-license-fee>

B. MANUSCRIPT ORGANIZATION AND FORMATTING:

spreadsheets for the main figures of the manuscript. If you would like to add source data, we would welcome one PDF/Excel-file per figure for this information. These files will be linked as supplementary "Source Data" files.

Corrections made in the revised manuscript

1. Amend the text to acknowledge the main limitation raised by Reviewer 1, that the effects observed under RNAi conditions may reflect a reduction in the number of mature oocytes.

We added this caveat in the results section and in the discussion:

“Thus, IFET-1 regulates levels of meiotic spindle and microtubule-related proteins in a post-transcriptional manner. **It is possible that these proteomic changes are, in part, due to defects in the production of full-sized, mature oocytes, which has been reported for worms lacking functional IFET-1(Sengupta et al., 2013).** However, proteomic analysis also revealed lower MEI-1, AIR-1, DLI-1, and TBB-1 levels in *ifet-1(ΔCHD)* mutant females, which are defective for oocyte storage and not active oogenesis (Figure S3D).”

“Why loss of IFET-1 would lead to a decrease in global translation is still unclear. There are several caveats to consider. First, **for our puromycin and imaging experiments, we performed RNAi knockdown of *ifet-1* prior to oocyte storage; thus, we currently cannot rule out the possibility that failed translation of some proteins could be an indirect effect of improper oocyte formation.**”

2. Modify figures according to Point 1 by Reviewer 1

Reviewer 1 requested that we outline the oocytes in all pictures where IFET-1 is knocked down so that the reader can assess oocyte number and size. We have revised figures 1 and 4 accordingly.

3. Modify the text according to Point 4 by Reviewer 1.

Reviewer 1 wanted us to emphasize that the puromycin analysis was performed on whole worms. We revised the text to say:

“We then analyzed global levels of translation in whole worms using a SUnSET assay. Puromycin incorporation was reduced in worms containing arrested oocytes vs. active oocytes (Figure S3E); thus, reduced translation is a hallmark of oocyte storage in *C. elegans*. In worms containing stored oocytes, depletion of IFET-1 further reduced puromycin incorporation compared with controls (Figure 3F).”

May 20, 2025

RE: Life Science Alliance Manuscript #LSA-2025-03387-TR

Dr. Jeffrey B Woodruff
The University of Texas Southwestern Medical Center
Cell Biology
6000 Harry Hines Blvd
Dallas, TX 75235

Dear Dr. Woodruff,

Thank you for submitting your revised manuscript entitled "Condensate-forming eIF4ET ensures adequate levels of meiotic proteins to support oocyte storage" incorporating the minor changes we requested. We would be happy to publish your paper in Life Science Alliance pending final revisions necessary to meet our formatting guidelines.

- Please consult our manuscript preparation guidelines <https://www.life-science-alliance.org/manuscript-prep> and make sure your manuscript sections are in the correct order.
- The contributions selected for Orhan Bukulmez do not qualify them for authorship. Please either update the contributions in our system and the Author Contributions section of the manuscript, or let us know if the author needs to be removed (and added potentially to the acknowledgment section).
- Please use the [10 author names, et al.] format in your references (i.e., limit the author names to the first 10).
- Please add callouts for Figures 1F; 3C-E and S3A-B to your main manuscript text.
- Please include a Data Availability section and consider uploading transcriptomics and proteomics data to a public repository.

LSA now encourages authors to provide a 30-60 second video where the study is briefly explained. We will use these videos on social media to promote the published paper and the presenting author (for examples, see <https://docs.google.com/document/d/1-UWCfbE4pGcDdcgzcmiuJI2XMBJnxKYeqRvLLrLS08s/edit?usp=sharing>). Corresponding or first-authors are welcome to submit the video. Please submit only one video per manuscript. The video can be emailed to contact@life-science-alliance.org

A. FINAL FILES:

B. MANUSCRIPT ORGANIZATION AND FORMATTING:

per figure for this information. These files will be linked as supplementary "Source Data" files.

Sincerely,

May 21, 2025

RE: Life Science Alliance Manuscript #LSA-2025-03387-TRR

Dr. Jeffrey B Woodruff
The University of Texas Southwestern Medical Center
Cell Biology
6000 Harry Hines Blvd
Dallas, TX 75235

Dear Dr. Woodruff,

Thank you for submitting your Research Article entitled "Condensate-forming eIF4ET ensures adequate levels of meiotic proteins to support oocyte storage". It is a pleasure to let you know that your manuscript is now accepted for publication in Life Science Alliance. Congratulations on this interesting work.

DISTRIBUTION OF MATERIALS:

Again, congratulations on a very nice paper. I hope you found the review process to be constructive and are pleased with how the manuscript was handled editorially. We look forward to future exciting submissions from your lab.

Sincerely,
